



# Development and implementation of SOMA: A Secondary Organic Module for Aerosol integration in high-resolution air quality simulations

Giannis Ioannidis[1], Nikoletta Bouloti[1], Paul Tremper[2], Chaofan Li[2], Christos Boikos[1], Nikolaos Rapkos[1], Till Riedel[2], Miikka Dal Maso[3], Leonidas Ntziachristos[1*]

[1]Mechanical Engineering Department, Aristotle University of Thessaloniki, GR 54124, Thessaloniki, Greece
[2]Karlsruhe Institute of Technology (KIT), TECO/Pervasive Computing Systems, 76131 Karlsruhe, Germany
[3]Aerosol Physics Laboratory, Physics Unit, Faculty of Engineering and Natural Sciences, Tampere University, Tampere, Finland

*Correspondence to*: L. Ntziachristos (leon@auth.gr)

**Abstract.** Secondary Organic Aerosols (SOAs) are formed following oxidation of Volatile Organic Compounds (VOCs) in the atmosphere and have a significant contribution to fine particulate matter concentrations. Understanding SOA formation is crucial, particularly in urban environments, where various emission sources contribute across different time scales. To decipher SOA formation dynamics, this study introduces SOMA (Secondary Organic Module for Aerosol) embedded in air quality modelling. SOMA considers VOC oxidation with OH using species concentrations, exposure duration, $NO_x$ levels and SOA yields as inputs, the latter obtained from the GECKO-A model. A total of 113 experiments are gathered from literature, involving four different VOC species (α-pinene, isoprene, limonene, and toluene), to produce correction factors depending on ozone ($O_3$) levels, relative humidity (RH), and temperature (T). SOMA was linked to CFD modelling and was used to characterise the dispersion of toluene SOA emissions from traffic in a heavily trafficked area in Augsburg, Germany. The dispersion model was used to simulate pollutant recirculation in the examined area using a novel approach by combining both local road traffic emissions and background sources. SOA formation from toluene was examined over a 12-h period. The results indicated that background SOA constituted 21-53% of the identified SOA mass. After 7 hours, the influence of background SOA on modelled concentrations became negligible due to precursor consumption and dilution. The combination of high-resolution pollution maps generated by CFD and atmospheric chemistry involving SOA formation enhances the air quality modelling capabilities and can provide valuable information to the scientific community.

## 1 Introduction

Secondary organic aerosols (SOAs) are formed after the oxidation of Volatile Organic Compounds (VOCs). Oxidation is a fundamental chemical reaction involving the transfer of electrons between reactants. In the context of SOA formation, oxidation primarily affects VOCs – the organic gas-phase precursors of SOAs (Srivastava et al., 2022). The specific oxidation reactions can be quite complex, involving multiple substances and are often driven by highly reactive free radicals,





like OH, $O_3$ and $NO_3$. Hydroxyl radicals (OH) have a great influence on SOA production as they are the most potent oxidant in the atmosphere (Saiz-Lopez et al., 2017). OHs react with VOCs, initiating chemical transformations that reduce VOC volatility and promote their condensation onto existing particles or formation of new ones (Hallquist et al., 2009). SOAs are a major component of fine particulate matter (PM), posing significant health and climate concerns (Ju et al., 2022; Lelieveld

et al., 2015). Their fine size allows them to infiltrate deep into the human respiratory system, and their complex chemical composition can trigger various health problems (Pope & Dockery, 2006). The presence of SOAs in the atmosphere can also influence climate by absorbing sunlight affecting the Earth's radiation balance (J. Li et al., 2022). Deciphering SOA formation is essential for effective pollution control strategies.

        To understand the nature and sources of SOAs, source apportionment studies play a crucial role due to the vast

diversity of VOCs emitted from both anthropogenic and biogenic sources (Claeys & Maenhaut, 2021; Dal Maso et al., 2016). These studies quantify the contribution of SOAs to PM concentrations and identify their origin. Studies conducted in urban environments revealed that SOAs constitute up to 40% of $PM_1$ (Brines et al., 2019; Carlton et al., 2009). To quantify the amount of SOAs formed under controlled conditions, researchers utilize specially designed oxidation chambers (Ahmad et al., 2017; Xu et al., 2014) and oxidation flow reactors (OFRs) (K. Li et al., 2019; Simonen et al., 2017). The oxidation

chambers allow researchers to isolate and investigate the influence of specific parameters, such as relative humidity (RH), $NO_x$ concentration, and OH levels, on SOA formation from various VOCs. While chamber studies provide valuable insights into the factors affecting SOA formation, it is crucial to acknowledge their limitations as these typically address conditions that may not be fully representative of real-world environments. On the other hand, OFRs simulate atmospheric oxidation conditions by exposing VOCs to oxidants such as OH radicals to study SOA formation but typically address short

timescales. A key metric in quantifying SOA used in these studies is SOA Yield, a dimensionless ratio calculated by dividing the mass of formed SOA by the mass of reacted VOCs. This factor quantifies the efficiency with which specific precursors contribute to SOA production.

        Several factors influence the formation of SOAs. Studies have shown a clear trend of increasing SOA yield with decreasing $NO_x$ concentrations (Brégonzio-Rozier et al., 2015; Ng et al., 2007). Branched structures of VOCs can further

reduce yield, particularly under conditions of high $NO_x$ concentrations (Lamkaddam et al., 2017; Loza et al., 2014). Temperature (T) also plays a role with cold conditions generally leading to higher SOA yields (Kim et al., 2012; Zhou et al., 2019). Relative humidity (RH) has an opposite effect, with higher humidity conditions often resulting to a decrease in SOA yields compared to drier environments (Jia & Xu, 2018; Zhang et al., 2015). It is important to note, however, that some studies suggest a minimal impact of humidity (Dommen et al., 2006; Nguyen et al., 2011).

SOA formation models are commonly used to simulate secondary organic aerosol production. One widely used approach is the Volatility Basis Set (VBS) model, which simulates the partitioning of semi-volatile and intermediate-volatility organic compounds (S/IVOCs) between the gas and aerosol phases (Sasidharan et al., 2023). However, the VBS approach does not include detailed chemical reactions. On the other hand, semi-explicit chemical mechanism models are employed, incorporating full chemical reactions for VOC oxidation and SOA formation (Eluri et al., 2018). The primary





limitation of this approach is its computational cost, which can be substantial due to the complexity of the chemical
     mechanisms involved. In this study, we develop SOMA (Secondary Organic Module for Aerosols), a computationally
     efficient SOA formation approach using an oxidation equation to estimate SOA generation from VOC oxidation. This model
     incorporates key variables such as $NO_x$ levels, SOA yields, oxidation duration, and hydroxyl radical (OH) concentrations
     specific to the environment being studied. The GECKO-A model is used for the calculation of the SOA yields used.
Additionally, correction factors based on experimental SOA formation data from literature are used to account for the effects
     of other influential factors, such as relative humidity (RH), and temperature (T) as well as other oxidants like ozone ($O_3$).
     The model is calibrated using experimental data allowing for greater accuracy while maintaining computational efficiency.

          Urban environments are hotspots for VOC emissions, originating from anthropogenic sources like traffic and
     households. These VOCs act as precursors to SOA and increase fine PM concentrations, adding to this health concerns
(Couvidat et al., 2013). Quantifying the dynamics of SOA formation in urban settings is a complex task due to the interplay
     between pre-existing SOA and unoxidized VOCs with SOA-forming potential. Studies have shown that using traditional
     chemical transport models accounting for primary VOC emissions can only quantify around 35% of observed SOA mass
     concentrations (Hodzic et al., 2009) and cannot provide high spatial resolution of SOA concentrations (Jiang et al., 2012).
     The residence time of some VOCs can range from hours to days, (Ehn et al., 2014), and the recirculation of these compounds
in urban environments can contribute to SOA production. No study has yet modelled and quantified the contribution to local
     SOA mass over various time scales in urban environments, highlighting a gap in current research. To address this challenge,
     we propose a combined modelling approach that uses SOMA in Computational Fluid Dynamics (CFD) for high resolution
     pollution estimations for a highly trafficked area of the city of Augsburg, Germany. CFD air quality modelling is a well-
     established approach for assessing pollution levels, particularly at street scale (Boikos et al., 2024; Du et al., 2021; Ioannidis
et al., 2024a; Jeanjean et al., 2017). Its strengths lie in the ability to model turbulent airflow, accurately represent the urban
     landscape, and provide high spatial resolution for pollutant concentration visualization and quantification. However,
     incorporating chemical processes in CFD modelling is highly challenging, and to date, no other study has included SOA
     chemistry in CFD modelling in urban environments.

          This study presents the development of SOMA, an SOA modelling tool, and its usage into dispersion modelling. It
seeks to enhance our understanding of SOA dynamics in urban environments. By embedding SOMA in realistic pollution
     data from simulations, we achieve high spatiotemporal predictions of SOA concentrations. Additionally, the combination of
     SOMA and the dispersion model allows for dynamically simulating SOA formation over various time scales, accounting for
     the recirculation of SOAs in the atmosphere through an innovative modelling approach. This modelling approach provides a
     comprehensive understanding of both local and background contributions to SOA levels in urban areas.






## 2 Methods

### 2.1 GECKO-A model

The GECKO-A model (Generator of Explicit Chemistry and Kinetics of Organics in the Atmosphere) is a chemical mechanism generator capable of handling a large number of species and reactions. It is supported by the National Center for

Atmospheric Research (NCAR) in the United States and the French Laboratoire Inter Universitaire des Systèmes Atmosphériques (LISA) (Aumont et al., 2005; Camredon et al., 2007). GECKO-A is designed to generate nearly explicit gas-phase oxidation mechanisms for one or more organic compounds under a range of atmospheric conditions. The model provides an online output library that serves as a comprehensive resource for researchers investigating SOA formation, offering data for over 50 different VOCs (Lannuque et al., 2018). A core capability of GECKO-A is its ability to calculate

SOA yields of various organic compounds under controlled conditions. The concept of SOA yield plays a critical role in understanding the formation of SOAs during atmospheric reactions. The SOA yield is a dimensionless quantity, expressed as a ratio, often calculated over a specific time interval. It essentially reflects the potential of a VOC species to produce secondary aerosol while it oxidizes. It is defined as the ratio of the mass of SOA mass formed (SOA) to the mass of the reacted VOC (Equation 1).

$Yield = \frac{SOA}{\Delta VOC}$                                                         (1)

The GECKO-A model produces SOA yields for five distinct $NO_x$ concentration levels, corresponding to remote (0.002 ppb), remote continental (0.0025 ppb), continental (5 ppb), polluted continental (20 ppb) and urban (200 ppb) environments. Users can select a specific VOC and a $NO_x$ pollution scenario, and then visualize how SOA yield changes over time, with temporal resolution ranging from seconds to hours. The SOA yields given by GECKO-A correspond to

constant conditions of temperature (T) at 25°C, ozone ($O_3$) concentration of 40 ppb and relative humidity RH at 70%.





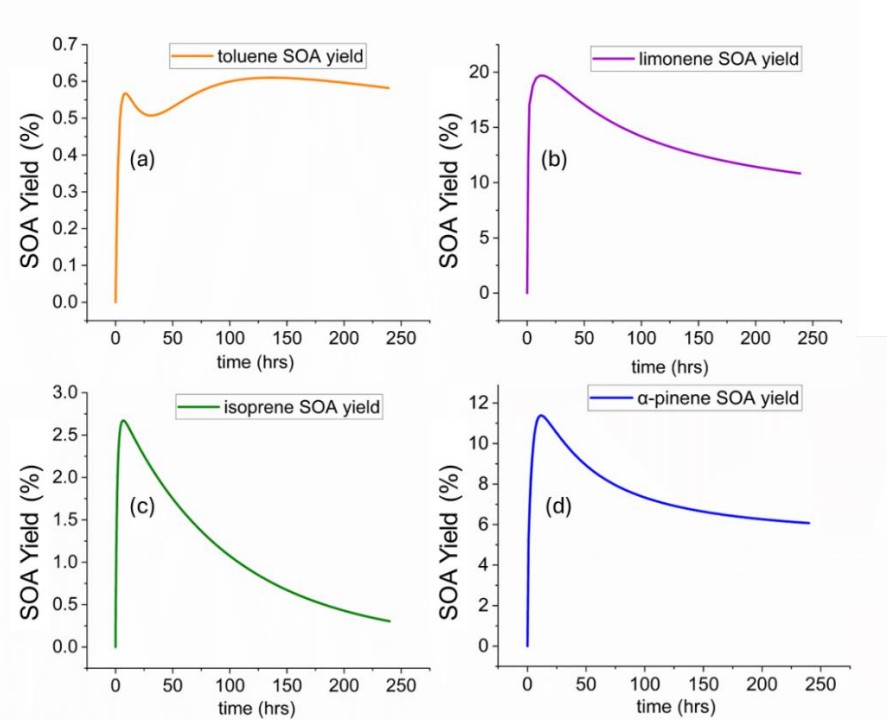

**Figure 1. Time series of SOA yields from GECKO-A for urban conditions (NOx=200 ppb), for toluene (a), limonene (b), isoprene (c) and α-pinene (d).**

Figure 1 shows trends in SOA yields from toluene ($C_7H_8$), limonene ($C_{10}H_{16}$), isoprene ($C_5H_8$) and α-pinene ($C_{10}H_{16}$)

under urban conditions ($NO_x = 200$ ppb) generated by GECKO-A, as examples. These compounds are selected due to their environmental significance and prevalence in experimental research studies (Ahmad et al., 2017; Bell et al., 2022; Brégonzio-Rozier et al., 2015; Oliva et al., 2023). For toluene (Figure 1a), SOA yield exhibits a sharp initial increase, stabilizing at around 0.6%, suggesting a balance between formation and loss processes influenced by $NO_x$ levels (Ng, Kroll, et al., 2007). In contrast, limonene (Figure 1b) and α-pinene (Figure 1d) show high initial yields (~20% and ~12%

respectively) that decrease over time, likely due to fragmentation of highly oxidized products (Cain et al., 2021). Isoprene (Figure 1c) presents a transient peak at 3%, declining rapidly to zero due to the quick fragmentation of its products, especially under high $NO_x$ conditions. These trends highlight the significant role of precursor reactivity and oxidation mechanisms in SOA formation and how SOA yields can vary through time.

A key limitation of predicted GECKO-A SOA yields is that parameters like RH, and T and oxidants like ozone are

held at reference values, which limits its use in real urban pollution simulations. Additionally, the model offers only five predefined $NO_x$ scenarios, limiting its flexibility in representing the wide range of atmospheric $NO_x$ concentrations observed in real-world environments.





## 2.2 Numerical model

This study uses OpenFOAM, an open-source CFD code, to model pollutant dispersion from traffic within the chosen urban
environment. Table 1 shows that the steady-state simpleFoam solver, employing the Reynolds-averaged Navier-Stokes
(RANS) approach, is used for calculating the velocity field (Boikos et al., 2024; Peralta et al., 2014; Rapkos et al., 2024).
The steady-state RANS approach is used because emission rates and meteorological conditions implemented as input to the
CFD model concern hourly values and are considered constant for every hour.

**Table 1. Numerical model information, equations and parameters.**

| CFD software | OpenFoam | |
|---|---|---|
| Modelling approach | RANS | |
| Turbulent model | k-epsilon | |
| RANS solver | simpleFoam | |
| Mass transport – advection diffusion equation | $\frac{\partial c}{\partial t} + \frac{\partial(\overline{u_j}c)}{\partial x_j} - \frac{\partial}{\partial x_j}\left(Dt\frac{\partial c}{\partial x_j}\right) = 0$ | (2) |
| Turbulent diffusion term (Dt) | $Dt = \frac{vt}{Sct}$ | (3) |
| Boundary layer velocity profile | $U(z) = \frac{u^*}{\kappa}ln\left(\frac{z+z_0}{z}\right)$ | (4) |
| Schmidt number (Sct) | 0.7 | |

The advection-diffusion equation (Equation 2) governs this transport process within the solver (Antoniou et al.,
2024; Miao et al., 2014). This equation includes the turbulent diffusion coefficient (Dt). Due to the dominance of turbulent
diffusion, particularly in low-wind speed scenarios, the molecular diffusion term (Dm) that is sometimes used in CFD
dispersion modelling can be safely neglected (Bonifacio et al., 2014; Rapkos et al., 2023). The turbulent Schmidt number
(Sct) is used to calculate Dt (Equation 3), with a recommended value of 0.7 (Tominaga & Stathopoulos, 2007). The k-ε
turbulence model is chosen for its well-known effectiveness in simulating pollutant dispersion within urban environments
(Pantusheva et al., 2022). The inclusion of Atmospheric Boundary Layer (ABL) profile (Equation 4) at the inlet boundaries
of the computational domain enhances the model's accuracy by capturing the characteristics of airflow within the urban
landscape (Yang et al., 2009). These profiles are determined using friction velocity (u*), Von Karman constant (κ), and
aerodynamic roughness length ($z_0$), providing a representative wind field behavior in urban settings.

## 2.3 Case study

This study focuses on Augsburg, Germany, a city in the Bavaria region. Figure 2a provides a broader view of the city, while
Figure 2b zooms in on the specific area chosen for the computational modelling. This area is selected for its central location
in the city. The case study area comprises of an official AQ monitoring station (Königsplatz KP) as shown in Figure 2b,
operated by the Bavarian State of Environment, strategically placed between two main roads at a height of 4m. It provides
concentration measurements of pollutants like $PM_{10}$, $NO_2$, NO, CO, benzene, toluene and xylene. On the southern side of the
study area, a background station (LFU) also provides pollutant concentration measurements for the same pollutants,




indicating the regional background levels of Augsburg city. On the southeast side of the urban area, a meteorological station provides information of wind speed and wind direction on an hourly basis.

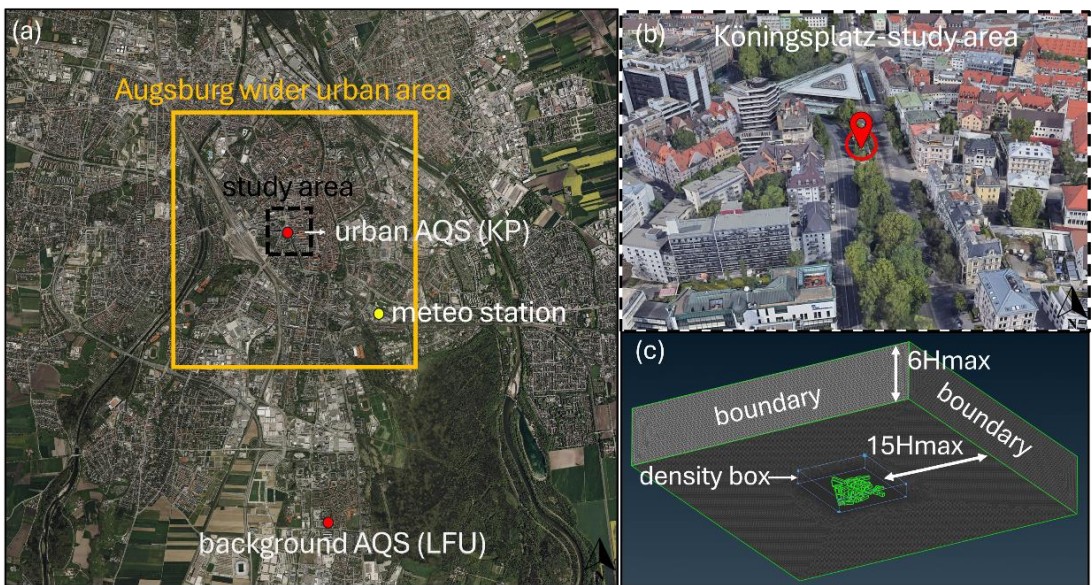

**Figure 2. Augsburg city area and focus on study area (a). Air Quality station placed on key position in the study area in Königsplatz (b). Computational domain and mesh development (c). © OpenStreetMap contributors 2024. Distributed under the Open Data Commons Open Database License (ODbL) v1.0.**

Representation of the urban environment's geometry is essential for reliable air quality modelling using Computational Fluid Dynamics (CFD). This allows for precise predictions of pollution levels within the urban canopy (Trindade da Silva et al., 2021). To achieve realistic representation of the urban environment examined we obtained the 3D geometry of the study area from Open Street Maps (OSM). Following data acquisition, the building surfaces were processed to remove inconsistencies and enhance their definition. As illustrated in Figure 2c, the digital domain extends beyond the immediate area. Table 2 explains that the distance between the study area and the computational domain's boundaries is set at 15 times the maximum building height (Hmax) and the height of the domain at 6Hmax, with the highest building in the area standing at 50m. This approach aligns with established practices (Blocken, 2015).

**Table 2. Computational domain and mesh information.**

| element | mesh resolution | dimension | length |
|---|---|---|---|
| emission sources | 0.25 m | City edges from boundaries | 750 m |
| buildings | 1 m | Length and width | 1800 m |
| boundaries | 15 m | height | 300 m |
| size box | 4 m | Hmax | 50 m |

A computational grid that covers the study area was constructed. The volume mesh that covers all the domain uses tetrahedral unstructured elements. The resolution on the buildings shown in Figure 2c is 1m, on the emissions sources 0.25m





and on the boundaries of the domain 15m (Table 2). Also, for high refinement around the buildings, a density box is
developed as displayed in Figure 2c that sets the maximum element size at 4m allowing for high spatial accuracy of the CFD
outputs with high refinement within street canyons and the sensor location. The computational grid consists of a total of 7
million elements. Mesh sensitivity analysis has been performed in previous studies performing CFD modelling in Augsburg
city (Gkirmpas et al., 2024; Ioannidis et al., 2024b), showing that this mesh resolution has acceptable convergence criteria
with scaled residuals within the accepted range of $10^{-6}-10^{-7}$ as proposed (Pantusheva et al., 2022).

## 2.4 Traffic emissions and meteorological conditions

The emission source setup (Figure 3a) shows that in the study are we included 5 unique road IDs. Hourly deviations from the
Annual Daily Traffic Volume (ADTV) for each road ID were produced for September 2018, resulting in a detailed hourly
traffic activity profile (Ioannidis, Li, et al., 2024a). VOC emissions were calculated using COPERT Street software for the
year 2018 in Germany based on the traffic activity. The vehicle classification considered: 73.6% passenger cars, 12.7% light
commercial vehicles, 7.5% motorcycles, 5.6% heavy-duty trucks, and 0.6% buses as taken from the Federal Motor Transport
Authority for the examined urban area in Augsburg. A total of 5 emission sources corresponding to the road IDs were
integrated within the digital model, which simulate emission rates from the road segments in the modelling domain.

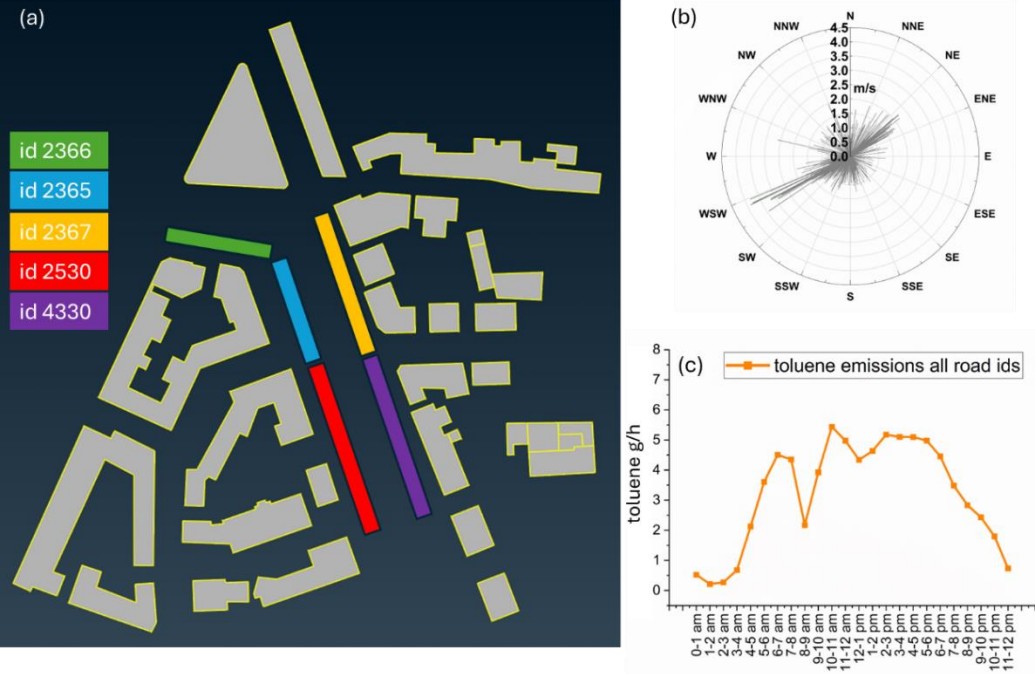

**Figure 3. Emission sources placed on road representing traffic activity (a). Rose graph of observed wind speed and direction
during September 2018 in Augsburg (b). Total toluene emissions from traffic from all the emission sources (c).**

Meteorological data collected for September 2018, when the traffic data was available, revealed a prevailing south-
westerly wind direction with an average speed of 0.90 m/s, with maximum wind speed observations at around 4 m/s as seen



in Figure 3b. The meteorological station that this data was obtained from is positioned on the southeastern side of the wider Augsburg area (Figure 2a), specifically chosen to minimize the influence of the urban landscape on wind observations. A 24-h period was chosen that exhibited a dominant south-western wind direction, with an average wind speed of 0.82 m/s, during 11/09/2018, to perform simulations for pollutant dispersion. This selection reflects the prevailing characteristics of the data collection period, ensuring that the results obtained are representative. Toluene is selected as the compound of interest, as this is primary emitted from traffic activity (Mehta et al., 2020) and its concentration measurements are available from the Air Quality station in our study area. Toluene emissions are estimated as a fraction of total VOC (Huang et al., 2020), calculated by the COPERT model. Figure 3c depicts the total emitted mass of toluene during the selected period for the 5 road IDs included in the model with distinguishable day and night emission profiles in the examined period, typical for urban driving volumes.

## 2.5 CFD Simulations using street and background sources

The CFD model developed for this study is used to produce pollutant concentrations from street and background sources. To save computational time, CFD simulations of distinct meteorological scenarios are performed, using 8 wind directions with an angle increment of 45°, for 2 sets of wind speeds, at 1 m/s (low) and 3 m/s (high). Running distinct wind scenarios to save on computational time has been effectively used to produce pollutant concentrations with CFD modelling in other studies (Boikos et al., 2025; Rivas et al., 2019). Simulations are performed for two different conditions. By only considering traffic emissions by the emission sources explained in Figure 3a, and by considering background concentrations on the boundaries of the computational domain, shown in Figure 2c. For traffic emissions, the concentration $C_{source,street}$ was set as initial condition in the CFD model setup on the grid corresponding to the emission sources specified in Figure 3a, calculated from emission releases in Figure 3c. For background sources, the initial concentration $C_{source,background}$ represents pollutant concentrations in neighbouring areas, set as boundary concentrations on the computational domain. By dividing the simulated concentrations in points of interest with the initial concentration at the boundaries or at the emission sources, we can determine the dimensionless value of the concentration ratio (CR) according to the distinct meteorological scenario examined (WD, WS). The concentration ratios for traffic and background contribution are defined in Equation 5.

$$CR_i = \frac{C_{point}}{C_{source,i}} \tag{5}$$

Where $i$ stands for 'source' or 'background'. To account for wind conditions not explicitly simulated in the CFD model, an interpolation function was used to estimate the concentration ratio for any observed wind speed ($WS_{real}$) and direction ($WD_{real}$), shown in Equation 6.

$$CR_{real,i} = f(WD_{real}, WS_{real}) \tag{6}$$



The pollutant concentration at any point of the domain can be calculated depending on the contribution of each source type. The pollutant concentration produced by the CFD model from street or background sources can be calculated using equation 7:

$$C_{point,i} = C_{source,i} \times CR_{real,i} \tag{7}$$

This methodology allows for the calculation of pollutant concentrations under varying meteorological conditions. A limited set of CFD simulations produces any pollution scenario using wind data based on real observations. For this work, we used a total of 32 sets of simulations (8 wind directions × 2 wind classes × 2 sources). The use of simulations of different pollution origins allows the identification and quantification of contributions from street and background sources. We name the interpolation functions as $fCFD_{street}$ and $fCFD_{background}$ from now on to represent the usage of the CFD model to calculate concentrations from street and background sources respectively on any selected point.

## 3. SOMA development

### 3.1 Module architecture

SOMA (Secondary Organic Module for Aerosol) calculates the formation of SOAs using a general approach based on the rate of an oxidation reaction (Equation 8). This equation follows first-order reaction kinetics and is commonly used in atmospheric chemistry to model how volatile organic compounds (VOCs) react with hydroxyl radicals (OH) (Chan et al., 2009). This equation incorporates the initial concentration of VOC and the rate of consumption, described through an exponential term that factors in the concentration of hydroxyl radicals [OH], the reaction rate constant between the OH radical and the precursor ($k_{OH}$) (Han et al., 2018; Singh & Li, 2007), and the oxidation period (Δt). The SOA yield (Y) of the specific compound for the given time is taken from GECKO-A.

$$SOA = [VOC] \cdot (1 - e^{-kOH \cdot [OH] \cdot \Delta t}) \cdot Y \tag{8}$$

This approach does not account for the influence of temperature and relative humidity, factors that affect the formation of SOA and other oxidants like ozone. To address this limitation and improve the model's reliability, we introduce a correction factor ($\lambda_{cor}$) into the equation (Equation 9). This factor is produced by collecting experimental information, as shown in Figure 14, for every investigated compound to determine the influence of other parameters ($O_3$, T, and RH), given that the yields taken from GECKO-A assume constant conditions with $O_{3gecko} = 40$ ppb, $T_{gecko} = 25°C$ and $RH_{gecko} = 70\%$ as explained in section 2.1. The parameterization of the new variables introduced is done by calibration with experimental data as will be elaborated in later section.

$$SOA = [VOC] \cdot (1 - e^{-kOH \cdot [OH] \cdot \Delta t}) \cdot Y \cdot (\lambda_{cor}) \tag{9}$$



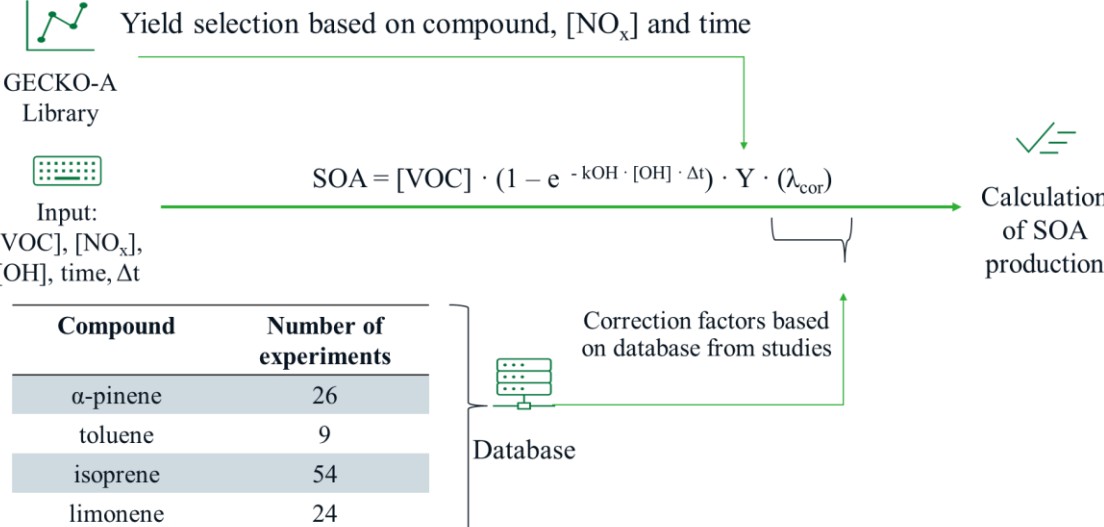


**Figure 4. SOMA uses [VOC], [NOx], [OH], time and Δt as an input to calculate the amount of SOA generated. SOA yields are selected from the GECKO-A output library. The model uses correction factors accounting for the influence of $O_3$, T and RH based on experiments**

### 3.2 Parametrization based on NOx concentrations for SOA yield selection

The five distinct $NO_x$ scenarios explained in section 2.1 can be used to account for SOA yields at any $NO_x$ level by interpolating between the data. Figure 5 shows the SOA yield for the four species examined at a specific time selected (time = 1h) for the five $NO_x$ scenarios. Figures 5a-d showcase high SOA yield at low-$NO_x$ levels, at remote (0.002 ppb) and remote continental (0.0025 ppb) $NO_x$ scenarios. Figures 2c and 2d show that the continental scenario (5 ppb) has higher SOA yield than the polluted continental (20 ppb), for limonene and α-pinene. Figures 2a and 2c show that the SOA yield is higher at the

polluted continental scenario compared to the continental one for toluene and isoprene, but with minimal differences. In all cases, the high-$NO_x$ scenario (200 ppb) demonstrates the lowest SOA yield.




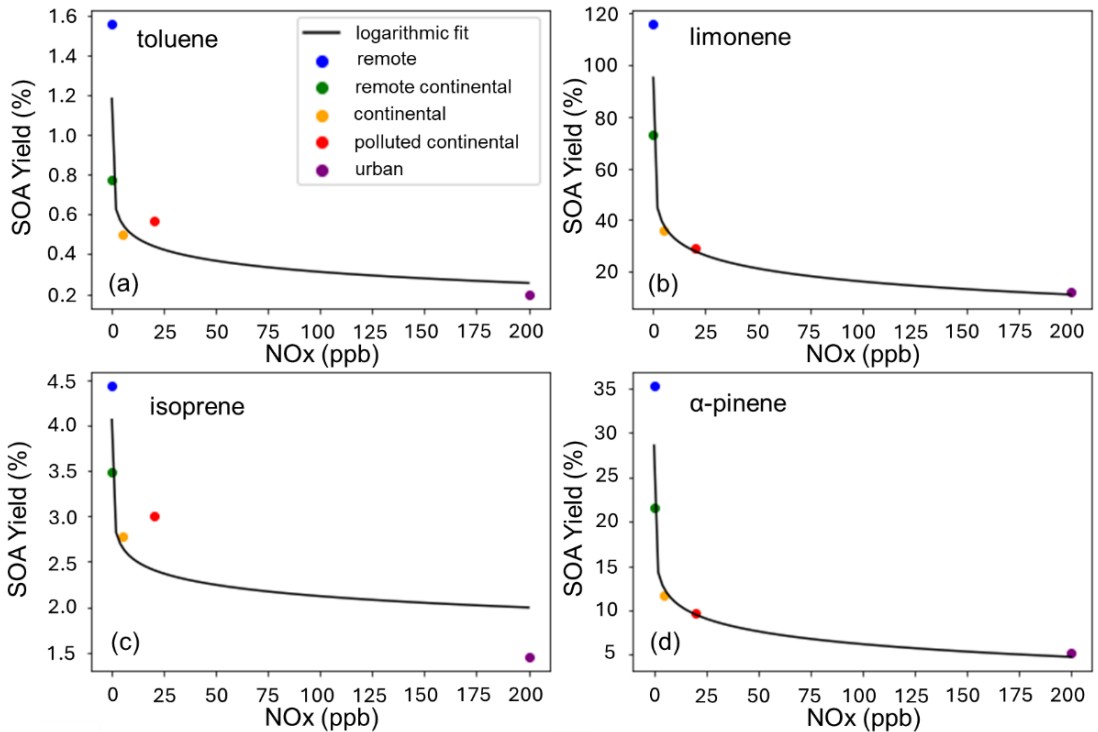

**Figure 5. SOA yields for 5 different NOx scenarios at 1 h, as generated by the GECKO-A model: remote (0.002 ppb), remote continental (0.0025ppb), continental (5ppb) polluted continental (20ppb) and urban (200ppb) for the four VOCs examined.**

The curve following the logarithmic fitting of the five points shows a decreasing trend with increasing $NO_x$ concentrations (Pullinen et al., 2020). At low-$NO_x$ environments, VOC oxidation typically proceeds through pathways that favor the formation of low-volatility compounds, which can readily partition into the aerosol phase, leading to higher SOA yields. In high-$NO_x$ conditions, the oxidative processes tend to shift towards pathways that result to higher-volatility products, thereby suppressing SOA formation. The fitting curve allows us to calculate the SOA yield of any given time-

period, for any $NO_x$ level.

### 3.3 Experimental data for SOMA calibration

To validate the SOA model and to provide reference for calibration, we gathered experimental information on the four mentioned VOCs in section 2.1. The experiments included information about the initial VOCs concentrations, duration of experiment and final SOA mass measured. Other experimental conditions like $NO_x$ concentration, ozone concentration, RH

levels, temperature, and OH levels are also documented, when available. Table 3 shows the data collected from two experimental studies, from 9 experimental measurements in total for toluene. In Deng et al. (2017) the three experiments collected had a reaction time of 5 h, $NO_x$ levels ranging from 68.2-72.2 ppb, RH from 50.5-55.5%, ozone levels from 42.7-77.6 ppb and temperature ranging from 25.4 to 26.9°C. In Chen et al. (2022) the reaction time of the experiments was 6 h,





RH was set at 5% and $NO_x$ levels ranged from 8-18.2 ppb. Temperature and RH were constant throughout the experiments at
T=26°C and RH=5%, and ozone levels ranged from 54.9-94.6 ppb. The data collected for every compound serve as reference for validation and as a base for experimental fitting based on the GECKO-A constant conditions to correct the SOA yields used by the SOMA. All experimental data for isoprene, α-pinene and limonene is available at the Supplementary Information (S.I.).

**Table 3. Experimental information from literature studies for SOA formation from toluene oxidation.**

| Study | Experiment number | VOC concentration [μg/m³] | T [°C] | time [s] | RH [%] | OH [molecules /cm³] | $NO_x$ [ppb] | $O_3$ [ppm] | SOA measured [μg/m³] |
|---|---|---|---|---|---|---|---|---|---|
| (Deng et al., 2017) | TOL1 | 298.95 | 25.8 | 18000 | 52.8 | 2.83E+6 | 72.2 | 0.0776 | 7 |
| | TOL2 | 233.15 | 25.9 | 18000 | 50.7 | 1780000 | 232.6 | 0.0076 | 2.1 |
| | TOL3 | 283.93 | 25.4 | 18000 | 50.5 | 1.56E+6 | 89 | 0.0427 | 0.8 |
| | TOL4 | 274.66 | 26.9 | 18000 | 55.5 | 2.94E+6 | 68.2 | 0.0683 | 6.2 |
| (Chen et al., 2022) | TOL5 | 54.79 | 26 | 21600 | 5 | N/A | 8 | 0.0696 | 0.17 |
| | TOL6 | 54.79 | 26 | 21600 | 5 | N/A | 2.8 | 0.0465 | 0.35 |
| | TOL7 | 197.79 | 26 | 21600 | 5 | N/A | 18.2 | 0.0946 | 5.8 |
| | TOL8 | 197.79 | 26 | 21600 | 5 | N/A | 8 | 0.0748 | 4.7 |
| | TOL9 | 197.79 | 26 | 21600 | 5 | N/A | 2.4 | 0.0549 | 3.2 |

### 3.4 Experimental fitting of SOMA

The developed SOA model generates the amount of SOA formed from oxidation of VOCs based on the chosen compound, the OH concentration, $k_{OH}$ and the selected yield (determined by $NO_x$ concentration and time). In the experimental fitting procedure, this process is repeated for every investigated compound, considering all the initial conditions from the experiments for the four compounds examined using the datasets explained in section 3.3. In the case that no information on
OH concentration could be found, we assumed [OH]=2×10⁶ molecules/cm³, typical of atmospheric conditions. The aim is to perform an analysis to determine the relationship between the ratio of $SOA_{exp}/SOA_{orig}$ and the ratios of $O_{3exp}/O_{3gecko}$, $T_{exp}/T_{gecko}$ and $RH_{exp}/RH_{gecko}$. The $SOA_{orig}$ is the value the SOA model calculated using the initial conditions of the experiments without the introduction of correction factors and the $SOA_{exp}$ is the value of SOA measurement given by each experiment. $O_{3exp}$, $T_{exp}$ and $RH_{exp}$ are the values taken from every experiment and the GECKO-A values are constant at
$O_{3gecko}$=40ppb, $T_{gecko}$=25°C and $RH_{gecko}$=70%. By incorporating the experimental data, the correction factor produced enables the adjustment of the selected yields to better reflect the influence of varying environmental conditions.





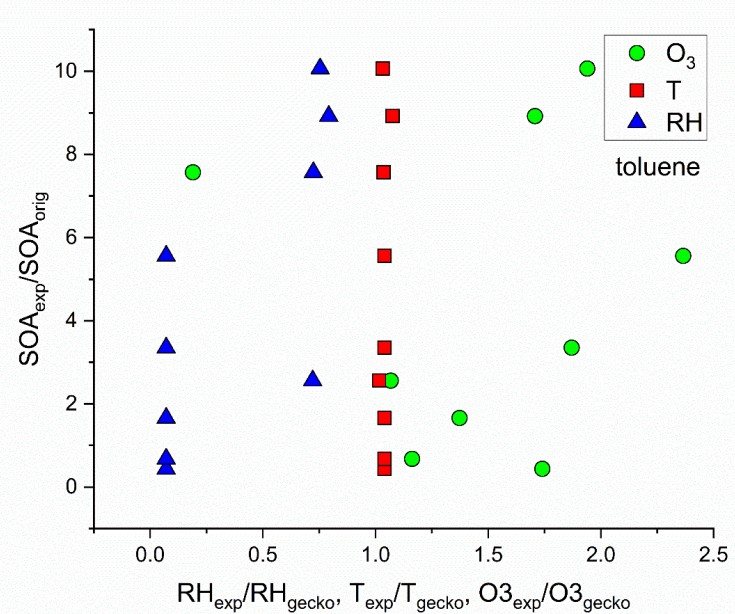

**Figure 6. Regression analysis of SOAorig/SOAexp with O3exp/O3gecko, Texp/Tgecko and RHexp/RHgecko based on the SOA formation experiments collected from literature for toluene.**

Figure 6 shows the regression graph between $SOA_{exp}/SOA_{orig}$ and $O_{3exp}/O_{3gecko}$, $T_{exp}/T_{gecko}$ and $RH_{exp}/RH_{gecko}$ for the case of toluene (Table 3). Figure 6 shows that for the case of $O_3$ the $O_{3exp}/O_{3gecko}$ ratios ranged from 1 to 2.4 with an exception at 0.2, indicating that the experiments collected examined higher ozone levels compared to GECKO-A ozone levels. The $T_{exp}/T_{gecko}$ ratios in all cases are very close to 1, as the temperatures examined in the experiments ranged from 25.8 to 26.9 °C, and the GECKO-A temperature is considered constant at 25 °C, showing that the effect of temperature is not

clear. The same analysis is performed for relative humidity (RH). The experimental RH used in the chamber studies was lower than the GECKO one with $RH_{exp}/RH_{gecko}$ ratios ranging from 0.05 to 0.8. To account for the influence of the 3 variables examined we performed multivariable linear regression analysis to produce equation 10 that describes the influence of the ratios of the variables affecting $SOA_{exp}/SOA_{orig}$. For every set of experiment, we use the ratios of the variables in equation 10, to produce the $SOA_{exp}/SOA_{orig}$ from the multivariable linear regression analysis. This ratio is then used to adjust

the modelled value closer to the experimental one by multiplying the $SOA_{orig}$ value.

$$\frac{SOA_{exp}}{SOA_{orig}} = -50.5 + 1.9 \times \frac{O_{3\,exp}}{O_{3\,gecko}} + 47.1 \times \frac{T_{exp}}{T_{gecko}} + 8.6 \times \frac{RH_{exp}}{RH_{gecko}} \qquad (10)$$

    Figure 7 shows the comparison of the original SOA model's outputs and the corrected ones after the use of correction factors based on the influence of $O_3$, T and RH, with the corresponding experimental SOA concentrations. In all cases, the $SOA_{exp}/SOA_{orig}$ ratio applied to the original model increased the calculated value, as it is evident that the original

SOA model tends to underestimate SOA formation using the GECKO-A toluene yields. The original SOA model underpredicted experimentally determined SOA mass by 76% on average. Using the correction factors a 6% under-





prediction across the average concentration is demonstrated in all studies. Taking into account all the corrections within the experimental fitting, we produced the $\lambda_{cor}$ as a correction factor used in SOMA to account for the influence of $O_3$, T and RH in SOA formation. This correction factor is applied on the SOA yield. The same analysis for limonene, isoprene and α-

pinene is shown in (S.I.).

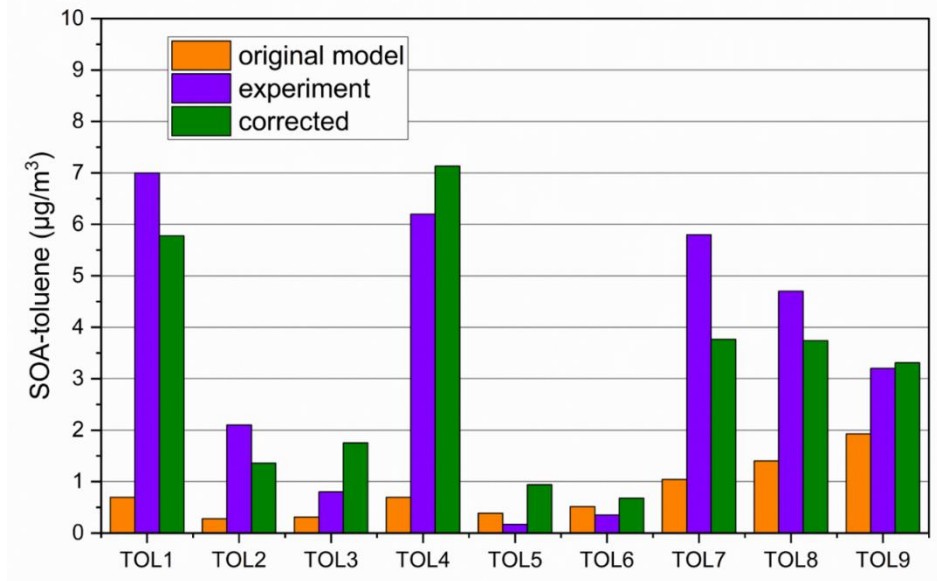

**Figure 7. Comparison between experimental SOA concentrations, original GECKO-A estimates and SOMA predictions in the case of toluene. TOL1-9 represent the experiments of SOA formation from toluene from Table 3.**

**3.5 Demonstration of NOx and OH influence on SOA formation**

After calibration, SOMA can be used to explore the impact of different parameters on SOA formation. Figure 8 shows the influence of OH and $NO_x$ on SOA formation within 1 h exposure of 1 μg/m³ for toluene in typical atmospheric conditions. Figure 8a demonstrates that within the range of specific OH levels ($10^5$ to $10^7$ molecules/cm³), the toluene SOA mass increases with increasing OH concentration (Sarrafzadeh et al., 2016). Beyond $5 \times 10^6$ OH molecules/cm³, the SOA mass reaches a plateau, indicating a maximum yield. Once OH radicals have oxidized all the precursors contributing to SOA

formation, the remaining unoxidized precursors can no longer participate in SOA mass growth, because of the insufficient availability of oxidizing agents (Song et al., 2019). In Figure 8b, $NO_x$ concentration exhibits an inverse relationship with SOA formation, aligning with the trends observed in the yield selection process from GECKO-A based on compound, $NO_x$ concentration and time described in section 3.2 (Figure 5). Figure 8b reveals that within the range of 0.01 ppb to 500 ppb $NO_x$, SOA mass is inversely proportional to $NO_x$. This behavior can be attributed to the scavenging of radicals at high $NO_x$

concentrations, thereby limiting oxidation and subsequent conversion of gas-phase precursors to the aerosol phase (Pullinen et al., 2020).





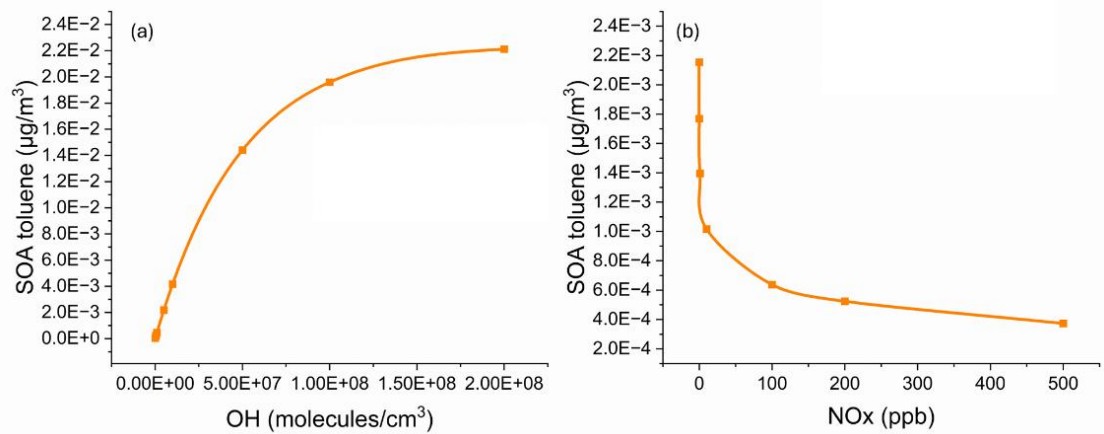

**Figure 8. Influence of (a) OH and (b) NOx concentrations on toluene-to-SOA formation, demonstrated by the SOMA model.**

This approach for predicting SOA formation under specific environmental conditions shows strong potential for Air
Quality (AQ) modelling applications. Estimated VOC concentrations can be input directly into SOMA. Using the VOC
emission source data, the model can set the emission duration as the oxidation period. To reflect environmental influence,
the model can also incorporate as an input $NO_x$ and OH concentrations, whether measured or modelled. Additionally, the
correction factors make the module more robust by accounting for other oxidants, like $O_3$, as well as environmental
parameters that affect SOA formation, such as relative humidity (RH) and temperature (T).

**4. Results**

**4.1 CFD model validation**

The validation of the CFD model was achieved by comparing simulated toluene concentrations from street emission sources
with measurements obtained from the KP urban traffic hotspot station. Toluene was chosen because it is the only SOA
precursor that can be directly linked to traffic in a city environment, and for which actual concentration measurements were
available. Following the methodology explained in section 2.5 by using traffic emissions and meteorological data for the
selected daily period, we produced gaseous toluene concentrations at the point of the Air Quality station, within the digital
domain. CFD concentration predictions were incrementally added to toluene concentrations measured at an air quality
station outside of the city (LFU station - Figure 2a), that served as the background. This approach assumes a uniform
background concentration within the modelled domain to account for the impact of sources not related to urban emissions
(Tchepel et al., 2010). Figure 9 demonstrates good agreement between measured and estimated toluene concentrations over a
24-hour period, with an average deviation of 8% and a correlation coefficient of 0.83. Figure 9 also depicts the $NO_x$
concentration measured during the chosen period at the KP station, showcasing the highest peak around 06:00-08:00



corresponding to the morning rush hour associated with traffic activity, the primary source of NO$_x$. Hourly NO$_x$ levels within the investigated area were used as input for the SOMA to calculate the corresponding SOA yield, as detailed in section 3.2.

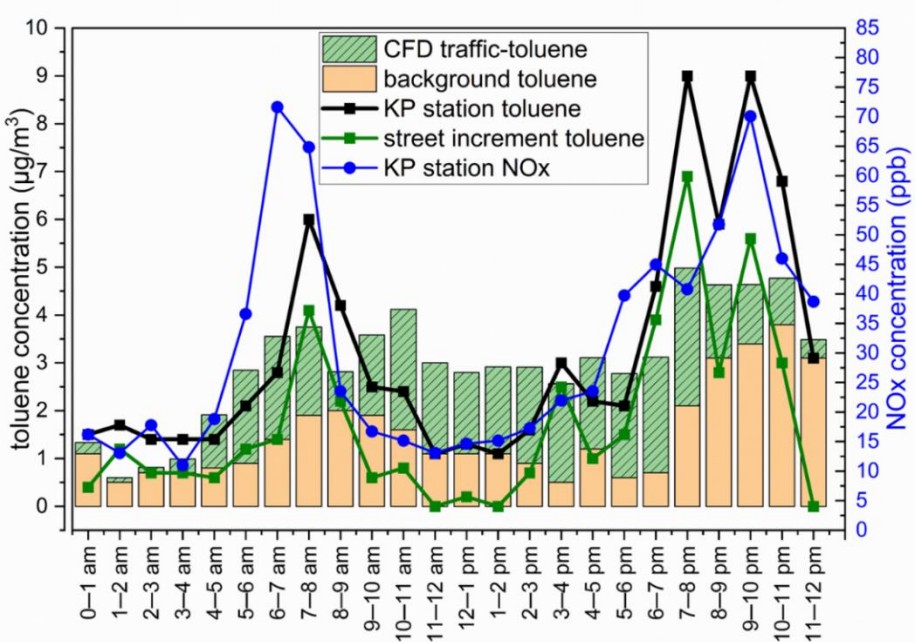


**Figure 9. Comparison of measured and estimated toluene concentrations at the location of the air quality sensor, together with measured NOx concentration.**

To directly compare CFD generated toluene concentrations with measurements, we identified the part of the measured toluene concentration that could be attributed to traffic activity. By subtracting the corresponding toluene concentrations

measured at the background station (LFU) located on the southern side of the urban area for the same hour from the measurement at KP, we can determine the street increment of the measurement. This is indicated by the green curve in Figure 9. The CFD generated toluene concentrations over the day, as shown by the stripped green bars. These exhibit distinguishable peaks during morning and afternoon periods. The same trend can be seen following the green curve of the street increment of measured toluene. The average daily concentration of street increment during this period was 1.75 μg/m³.

When using only traffic emissions as input, the CFD model produced average daily toluene concentration of 1.50 μg/m³, closely matching the observed street increment, producing values within the same range as measurements. While the hourly comparison showed some negative increments—due to instances when the background measurement was higher than the KP station measurement (between 11−12 am and 1−2 pm, as shown in Figure 8)—the daily concentrations obtained from both the CFD model, and the street increment showed a deviation of 15%. The agreement with measured toluene concentrations

suggests that the model effectively replicates the dispersion of traffic-related toluene within the urban environment, especially if one considers uncertainties of exact traffic rates and emission factors related to the calculation of total emission rates.





### 4.2 SOA estimates

#### 4.2.1 Coupling SOMA with CFD

The temporal resolution of available wind data and emission rates was hourly, so CFD derived concentrations were also of hourly resolution. In urban environments, the OH levels vary between $0.7 - 8{\times}10^6$ molecules/cm$^3$ depending on light intensity, temperature and the presence of certain pollutants (Cho et al., 2023). No information on urban OH levels was available for this study, so we selected an OH concentration of $2{\times}10^6$ molecules/cm$^3$. VOC to SOA conversion was predicted by means of eq.9. where VOC concentrations were produced by the CFD for the different cells of the domain, assuming only
toluene.

#### 4.2.2 Pollutant recirculation approach

    Certain VOCs can remain in the air for hours to days, allowing them to recirculate in urban areas and contribute to SOA production. This contribution often originates from neighboring areas and can accumulate over various time scales. To estimate the contribution of SOA originating from neighboring areas to the study domain, we assume that traffic patterns are
consistent throughout the broader urban region due to similar driving conditions of urban environments. By focusing on our specific case-study area, we argue that traffic-related emissions can be considered representative of the surrounding zones, given the study area's central location within the city of Augsburg (Figure 2a). SOA from neighboring areas could manifest itself in the modelling domain in two ways. First, as transported SOA formed in previous time frames outside of the modelling domain and getting diluted through time, as it is transported to the modelling domain. Second, from previously
emitted VOC that is transported to the modelling domain and forms fresh SOA within the domain of interest. To clarify, SOA in the modelling domain can have three different origins:

1. Freshly formed SOA resulting from VOC emissions produced within the domain during each hour modelled. This is referred to as fresh SOA **(F-SOA)**.
2. Previously formed SOA outside of the modelling domain, transported into the study area within the given hourly
time frame, referred to as background SOA **(BG-SOA)**.
3. Freshly formed SOA resulting from VOC transported into the modelling domain from neighboring areas, referred to as background fresh SOA **(BGF-SOA)**.

    The sum of BG-SOA and BGF-SOA is denoted as background SOA and the F-SOA is denoted as local SOA in the next sections. Equation 11 is used to calculate the total SOA mass concentration.

$SOA\ total = (F - SOA) + (BG - SOA) + (BGF - SOA)$                       (11)



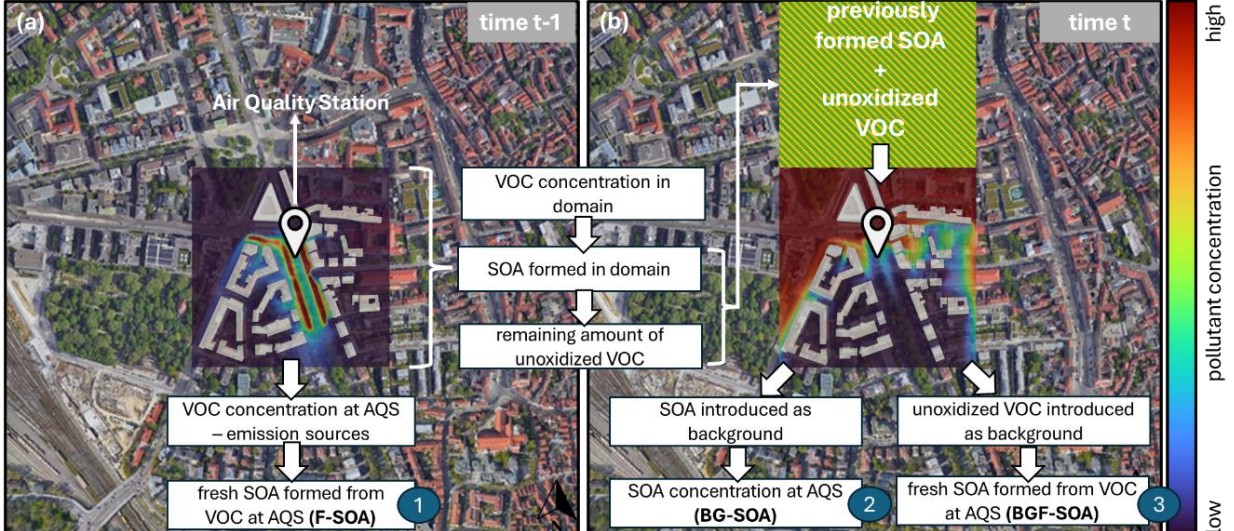

**Figure 10. Schematic representation of the approach for identifying fresh SOA (F-SOA) (a), background SOA (BG-SOA) and background fresh SOA (BGF-SOA) (b). © OpenStreetMap contributors 2024. Distributed under the Open Data Commons Open Database License (ODbL) v1.0.**

The representation of the approach used to calculate local and background contribution to simulated SOA concentrations is shown in Figure 10. Figure 10a shows that fresh SOA (F-SOA) for each time period is calculated using VOC concentrations generated by the CFD street emission modelling. For each hourly period, the CFD model estimates VOC concentrations produced by local street emissions and SOMA estimates F-SOA concentrations.

    By using the CFD generated toluene concentration fields from street emissions, we can also calculate the average

VOC concentration in the whole domain, by calculating the concentration of every point of the grid. We then implement the domain VOC concentration in SOMA, to calculate the SOA formed in our case study area (equation 12) and VOC consumption given by the term ΔVOC (equation 13). The remaining unoxidized VOC ($VOC_{remaining-domain}$) amount after can then be calculated using equation 14.

$$SOA_{domain}t-1 = fSOMA(VOC_{domain}t-1) \tag{12}$$

$$\Delta VOC_{domain}t-1 = [VOC_{domain}t-1] \cdot (1 - e^{-kOH \cdot [OH] \cdot \Delta t}) \tag{13}$$

$$VOC_{remaining-domain}t-1 = VOC_{domain}t-1 - \Delta VOC_{domain}t-1 \tag{14}$$

    The terms $SOA_{domain}t-1$ and $VOC_{remaining-domain}t-1$ represent the average SOA formed and remaining VOC in the domain respectively as calculated by equations 12 and 14 for time (t-1). These two terms are then used as input to the CFD simulations of background sources for the next time period (t) as illustrated in Figure 10b. They represent pollutants present

in the domain of interest from neighbouring areas, assuming similar traffic activity in both environments. The CFD background model calculates the concentration of already formed SOA at the point of KP station introduced as background, based on the meteorological conditions of time t. The concentration calculated is denoted as BG-SOA.




BG-SOA$_t$= $fCFD_{background}$(SOA$_{domain}$t)                                           (15)

The CFD dispersion model is then used to calculate the VOC concentration at the point of KP station, from

background remaining unoxidized VOC. The VOC$_t$ concentration can be calculated by equation 16, using meteorological

conditions of the corresponding period.

VOC$_t$=$fCFD_{background}$(VOC$_{remaining-domain}$t)                                        (16)

The SOA formed from VOC presence during that time can be calculated by equation 17, using the environmental

conditions of that time. This is now denoted as background fresh SOA (BGF-SOA).

BGF-SOA$_t$= $fSOMA$(VOC$_t$)                                                   (17)

By repeatedly calculating the formation of SOA and consumption of the VOC we can use the combination of the

SOA model and the CFD to calculate for every time period the amount of F-SOA, BG-SOA and BGF-SOA. At the end the

total SOA concentrated at the point of interest, will be the sum of the three different SOAs calculated (equation 11). The

freshly produced SOA from street emissions at the corresponding time (F-SOA), the already formed SOA that is transported

and diluted (BG-SOA) and the freshly produced SOA from background VOCs (BGF-SOA), simulating a pollutant

recirculation effect within the urban environment.

**4.3 Influence of local and background contribution to SOA concentrations using the pollutant recirculation approach**

To demonstrate the model's application in accounting for SOA formation, we selected a 12-hour period (06:00−18:00) on the

day examined, considering that sunlight is present during this period to initiate the photo-oxidation of toluene. SOA is

mostly formed during periods with UV radiation to facilitate the formation of hydroxyl radicals (OH) via sunlight exposure

(Altshuller, 1989; Wang et al., 2021) as they have a critical role on SOA formation.

In Figure 11, each hour shows the contribution of SOA from toluene emissions, starting from the initial analysis at

06:00. For instance, SOA formed from toluene emissions from traffic between 07:00 and 08:00 makes up 79% of the SOA

mass during that time, representing fresh SOA (F-SOA) formed from recent emissions. The remaining 21% of the SOA mass

has a background origin. This background portion includes SOA that had already formed from toluene emissions between

06:00 and 07:00 and has since diluted (BG-SOA), as well as additional SOA formed between 07:00 and 08:00 from toluene

emitted during the earlier 06:00–07:00 period (BGF-SOA). Later in the day, for example, between 17:00 and 18:00, 47% of

the SOA is produced locally from toluene emissions during that specific time, while the remaining 53% has background

origins, resulting from emissions that occurred from 06:00 up until 17:00. Notably, less than 1% of the SOA concentration

during this time can be attributed to emissions before 11:00–12:00, indicating minimal contributions from earlier emissions.

The trend shown in Figure 11 indicates that SOA contributions from toluene emissions at 06:00–07:00 steadily decline

throughout the day until 18:00. This decline results from the progressive consumption and dilution of the toluene responsible

for SOA formation, as estimated by the recirculation approach. Similar behavior is observed for toluene emissions at other

times, where their contribution to SOA mass decreases through time.





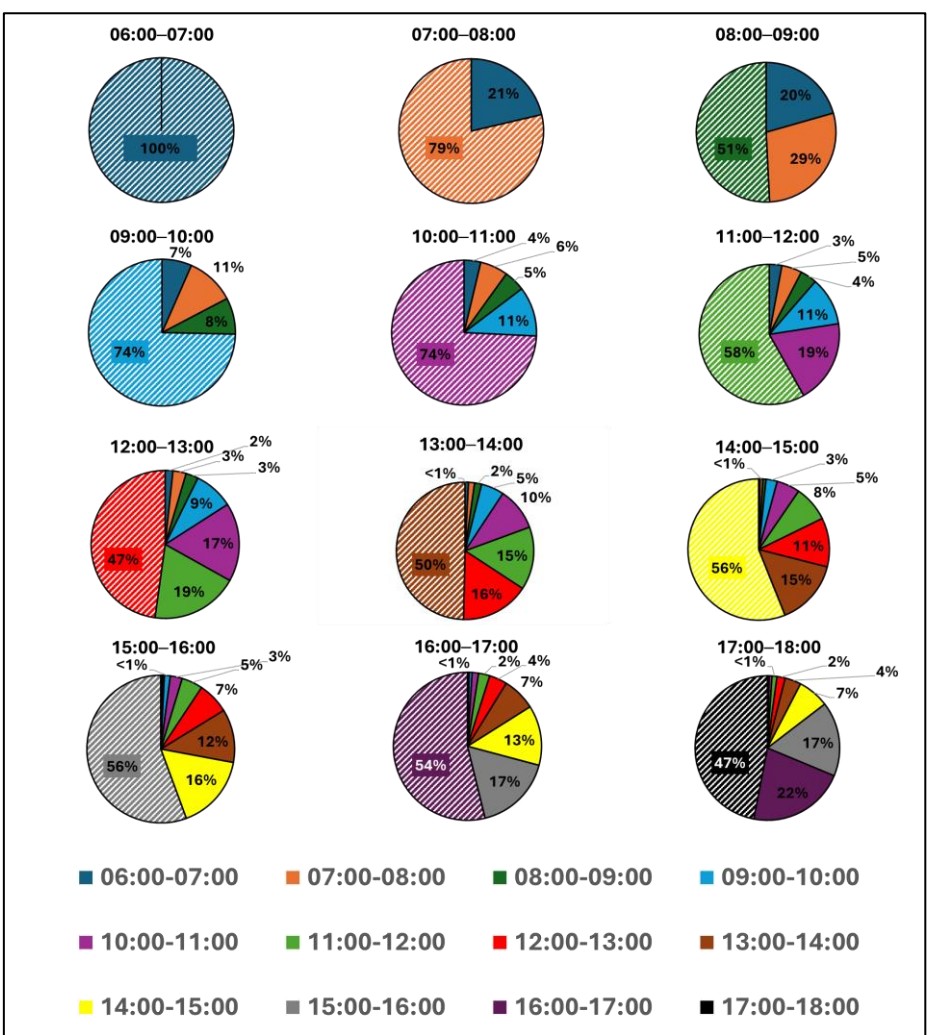

**Figure 11. Local (stripped slices) and background contribution on SOA concentration from traffic-toluene emissions produced by recirculation approach.**

This analysis allows us to assess the cumulative contribution of SOA formed over different timescales, helping to identify when older SOA formations have a negligible impact on current SOA concentrations. In Figure 12, the SOA concentration at time=0 represents locally formed, fresh SOA (F-SOA) for each period examined. For any given period, the total SOA originating from time=0 corresponds solely to the freshly formed SOA. As time progresses, the total SOA in each period increases as SOA from prior time intervals is incorporated, adding background-originated SOA mass to the overall modelled SOA. In Figure 12, the black curve illustrates the cumulative SOA concentration for the 17:00–18:00 period. Here, SOA originating specifically from 17:00–18:00 (i.e., time=0 for this period) has a concentration of $1.53 \times 10^{-3}$ µg/m³, representing the local SOA formed during this time. Moving one step backward along the x-axis to time=-1 (16:00–17:00), the total SOA concentration increases to $2.22 \times 10^{-3}$ µg/m³ as SOA from the prior hour is added. This approach highlights





how cumulative SOA concentrations evolve over time, illustrating how contributions from earlier time periods influence total SOA levels across examined timescales.

By continuing to account for the background SOA contribution to the total SOA mass concentration calculated, we can determine at what point this cumulation ceases to make significant difference to the total SOA concentration. Figure 12 reveals that the contributions of SOA stabilize after 7 hours, indicating that earlier background SOA generated by traffic has a minimal impact on current concentrations. This stabilization occurs because the volatile organic compounds (VOCs) emitted in earlier time periods have either been consumed through chemical reactions or broadly diluted in the atmosphere.

As a result, after 7 hours, the cumulative effect of background SOA becomes negligible.

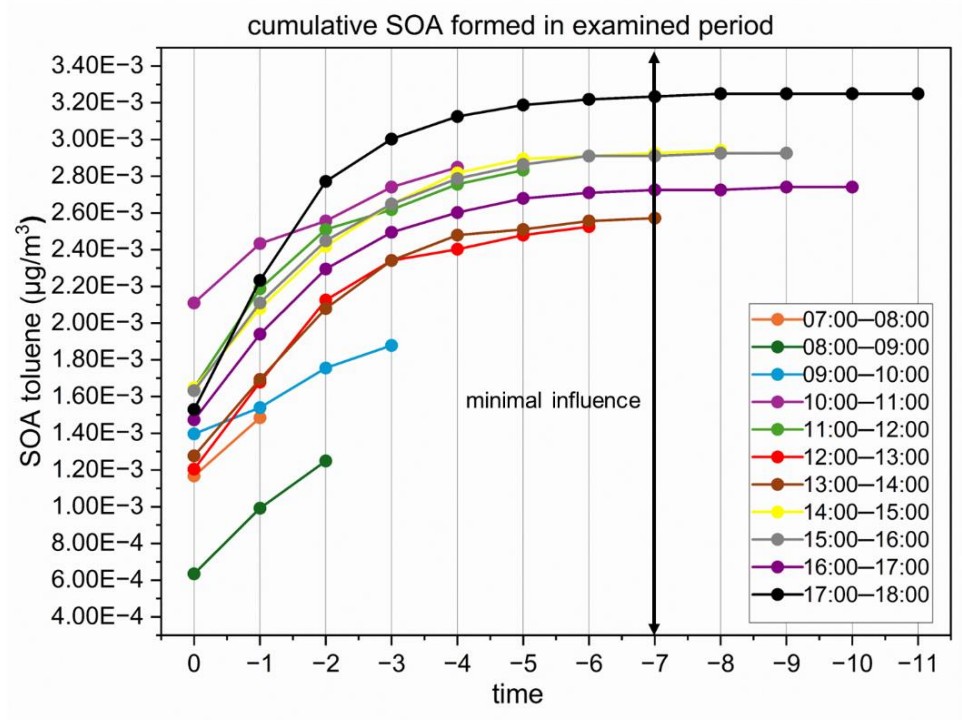

**Figure 12. Cumulative SOA concentration from traffic-toluene at the sensor location. The x-axis indicates how many hours ago the SOAs were formed and have an influence on SOA concentrations in every time-period.**

       Our approach shows that although local contributions remain more dominant overall, the impact of total

background SOA grows stronger over time due to the accumulation of previously formed SOA mass from multiple time scales. The findings of this work are specific to toluene, the VOC examined in this modelling approach. It is crucial to recognize that different VOCs may exhibit unique behaviors, and the background SOA from their oxidation could yield different results. Therefore, the conclusions drawn for toluene may not be directly applicable to other compounds, and each precursor's contribution to background SOA should be individually assessed. Nevertheless, toluene is a significant

component of traffic emissions, making these conclusions broadly representative. Future research could expand this modelling approach by including various other primary emitted precursors to better understand the local and background



effects on SOA formation over different timescales from dominant urban emission sources. Such insights are essential for developing effective policy measures to mitigate these pollutants, as they cannot be accurately quantified by simply correlating SOA presence with primary emissions.

## 5 Conclusions

Secondary Organic Aerosols (SOAs) are formed by the oxidation of Volatile Organic Compounds (VOCs) and have an important contribution to fine PM concentrations. Understanding SOA formation is crucial as the high residence time of some VOCs, that ranges from hours to days, can contribute to SOA production. To capture the dynamic effect of SOA formation from traffic activity in urban areas we introduce Secondary Organic Module for Aerosol (SOMA), a module that can be implemented in air quality modelling. SOMA uses an oxidation equation based on OH radical interactions with precursors and incorporates data such as VOC, OH and $NO_x$ concentrations, oxidation duration, and specific oxidation rate constants. By analyzing 113 chamber experiments involving α-pinene, isoprene, limonene, and toluene, correction factors were established based on ozone ($O_3$), relative humidity (RH), and temperature (T) influence on SOA production. The correction reduced deviations from experimental values.

To illustrate SOMA implementation in air quality modelling, a CFD model is developed to simulate toluene emissions from traffic in a high-traffic area of Augsburg, Germany. Simulations were used to calculate toluene concentrations from both street and background sources. The results of these simulations were used in a novel approach, reflecting recirculating pollution from neighboring areas influenced by meteorological conditions. The modelled toluene concentrations were validated against official air quality station measurements. SOA formation from CFD-generated toluene was examined over a 12-hour period, considering atmospheric conditions and OH presence. A limitation of this study is that OH levels were assumed based on common urban conditions. Future implementation of SOMA in air quality models could benefit from actual OH trends observations in the examined environments for more representative results.

The modelling approach estimated the local and background effect on SOA formation from traffic in the examined area. SOMA and CFD coupling showed that SOA originating from toluene oxidation has diverse sources through time. During the 12-hour period examined, focusing on the point of the air quality station of the area, the background SOA was 21-53% of identified SOA mass. Also, analysis has shown that after 7 hours, the contribution of background SOA has negligible influence on the concentrations modelled. The recirculation effect of each precursor individually has strong influence up to 7 hours and decreases due to the consumption and the dilution both of the SOAs and the VOCs. The results of this integration are based on toluene, but it's important to acknowledge that other VOCs may exhibit different SOA yields. Future development of SOMA could expand to include the full spectrum of VOCs from specific emission sources.

This study provides valuable insights into the formation and sources of secondary organic aerosols (SOAs), offering guidance for the scientific community to better understand SOA production from volatile organic compound (VOC) emissions. As SOAs significantly affect air quality and human health, this research provides innovative modelling

techniques that enhance air quality capabilities with high-resolution spatiotemporal dispersion and chemical transformation
approaches. These methods provide a clearer picture of SOA formation dynamics in urban areas. Since SOAs cannot be
directly linked to specific sources and are not currently regulated, this study is crucial for uncovering their formation
processes and advancing knowledge in this area.

**Code and data availability**

The codes and datasets in this publication are available to the community, and they can be accessed by request to the
corresponding author.

**Author contributions**

Conceptualization, G.I., M.D.M. and L.N.; Methodology, G.I., N.B., N.R., C.B., M.D.M. and L.N; Software, G.I., N.R. and
C.B.; Investigation, G.I., N.B., N.R. and C.B.; Data curation, G.I., P.T., C.L. and T.R.; Writing – original draft preparation,
G.I., N.B. and L.N.; Writing – review & editing, G.I., M.D.M. and L.N.; Supervision, M.D.M and L.N.; All authors have
agreed to the version of the manuscript.

**Competing interests**

The contact author has declared that none of the authors has any competing interests.

**Funding**

The research was funded by the Helmholtz Association of German Research Centres, through the GRACE foundation under
funding number 51, in the frameworks of HEPTA project.

**Acknowledgements**

This study was conducted as part of the HEPTA project in the context of "Air Quality in Smart Cities" and is supported by
the GRACE organization. GRAZ university of technology provided traffic information and the KIT air quality data.
GECKO-A model results obtained from J. Lee-Taylor and C. Drews, Atmospheric Chemistry Observations and Modeling
(ACOM), NSF-NCAR, https://www.acom.ucar.edu/gecko/output-library.shtml, accessed 12/01/2022. By using the CFD
software we acknowledge OPENFOAM® as a registered trademark of OpenCFD Limited, producer and distributor of the
OpenFOAM software via www.openfoam.com. Maps available under the Open Database Licence from: openstreetmap.org
The authors used ChatGPT 3.5 only to improve the readability and language of this work.



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
