# Peer review of "Development and implementation of SOMA: A Secondary Organic Module for Aerosol integration in high-resolution air quality simulations"

_EGUsphere, 2025_

## Author Comment (AC1)

**Response to comments on "Development and implementation of SOMA: A Secondary Organic Module for Aerosol integration in high-resolution air quality simulations"**

We would like to thank the reviewers for evaluating our manuscript and for providing their constructive comments. Below you can find point-by-point answers to the RC1 reviewer's comments. All references are provided at the end.

**RC1: 'Comment on egusphere-2025-193', Anonymous Referee #1, 29 Jun 2025**

Ioannidis et al. develop parameterizations for SOA mass yields using GECKO and laboratory data and combine those parameterizations with a CFD model to examine the spatial and source patterns of SOA in a targeted urban area. SOA is a complex pollutant with a multitude of sources and pathways. It contributes to PM1 significantly and there is a need to develop approaches that can simulate SOA formation in atmospheric models. Hence, the paper's need is justified. However, the paper's motivation (described in the introduction) is adequate but not strong. I also found the methods inadequate and confusing. The use of CFD seems novel but I contend that the use case to study SOA is not a good fit. Overall, the paper is weak, and I do not recommend publication of this paper in ACP. The comments below outline my most important objections, and I would strongly recommend the authors consider those as they think of novel applications for their CFD modeling.

Major comments:

**Comment 1:**

1.    Even for highly reactive VOCs, the timescales for SOA production at ambient concentrations of OH and O3 are on the order of hours (even longer for species like toluene that has an e-folding lifetime of over 1.5 days). The motivation for using CFD in 'street canyons' to estimate SOA production – which is more regional - isn't quite strong. I remain highly skeptical of the primary finding that the SOA in a domain this small (1.8 km) is dominated by SOA from local sources within the domain. I suspect that CFD at this spatial scale would be much more useful in tracking the spatiotemporal evolution of primary particles and gases where transport and dilution are much more relevant than chemistry. In my opinion, to get at airshed level burdens of SOA, 0D box models (e.g., Hayes et al., ACP, 2015) and high-resolution chemical transport models (e.g., Pennington et al., ACP, 2021) are likely better tools to model the formation, evolution, and properties of SOA.

**Response:**
We appreciate the reviewer's concern regarding the spatial scale of SOA formation processes. We agree that SOA chemistry, especially for less reactive species like toluene, extends over larger spatial and temporal scales. However, the objective of our work is not to capture the full regional SOA burden, but to explore spatial heterogeneity and source attribution in urban hot-spots using a hybrid high-resolution approach. CFD is used not to simulate chemical

transformations directly, but to produce high-resolution VOC concentration fields and wind-dependent recirculation patterns that feed into SOMA.

To address the reviewer's concern more explicitly, we have revised the Introduction and Discussion sections to better position the purpose and limitations of using CFD in this context. In L94-96 we revised the manuscript to state: "While the full formation of SOA can span hours to days, initial oxidation steps begin within the first hours after VOC emission. In street canyon environments, turbulent recirculation can extend the residence time of reactive VOCs, enabling early-stage SOA formation within the local domain." Also, in L104-106 we explain further: "In this study, CFD is not used to explicitly simulate complex atmospheric chemistry, but rather to resolve realistic concentration and flow fields that feed into SOMA, which calculates SOA formation using corrected yield expressions."

**Comment 2:**

2. Figure 1: Details about the initial VOC, OH/O3 concentration, OA mass loading are all missing. Why are results shared in the methods section?

**Response:**
We thank the reviewer for this observation. Figure 1 displays the GECKO-A modelled SOA yields over 240 hours for four selected VOCs (toluene, limonene, isoprene, and α-pinene) under fixed conditions. These are not simulation results from our model but rather yield outputs from GECKO-A used as input data in SOMA. As such, we placed the figure in the Methods section to document the origin of the input data.

As stated in L133–135: "The SOA yields given by GECKO-A correspond to constant conditions of temperature (T) at 25°C, ozone ($O_3$) concentration of 40 ppb and relative humidity RH at 70% (Camredon et al., 2007)."

The initial mass mass loading is 1 pptvC, and this is now explicitly mentioned in the caption of Figure 1 in L136 for clarity:

[Figure]

Figure 1. Time series of SOA yields from GECKO-A for urban conditions ($NO_x$=200 ppb), for toluene (a), limonene (b), isoprene (c) and α-pinene (d) for an initial organic mass loading of 1 pptvC.

**Comment 3:**

3. Sections 3.2-3.4: I don't understand the rationale for these methods. I see several glaring problems. First, toluene SOA has been widely studied and there 10s (if not 100s) of studies that have documented SOA mass yields. While there isn't an expectation to include every published study on toluene SOA, what is expected is a rationale for why these (i.e., Deng and Chen) were picked and how they are representative of the broader consensus about toluene SOA mass yields. Second, it is unclear why GECKO wasn't directly run for the same experimental conditions for initial VOC, NOx, RH, and T. Depending on the model-measurement performance, a case could have been made for parameterizing SOA_exp/SOA_org to differences in O3. Regardless, I would still be skeptical of this parametrization as no attempt was made to mechanistically explain why the model underestimates the measurements so substantially. Third, SOA is expected to be a strong function of the OA/SOA mass loading (depending on whether an organic seed was used to aid SOA condensation) and hence any model-measurement difference in yield is likely to also be a function of the SOA mass concentration (which in itself is the primary output that is being used to compute the yield). This non-linearity is probably the most difficult to resolve.

**Response:**

We thank the reviewer for raising these important points. The selection of the (Deng et al., 2017) and (Chen et al., 2022) studies was based solely on data availability: these are experiments we found that reported the required input parameters (VOC concentration, RH, temperature, $NO_x$, ozone, OH, duration, and final SOA mass), which were necessary for our calibration analysis. Our goal was not to provide a comprehensive representation of toluene

SOA literature but to build a consistent dataset for regression-based correction. To highlight this we state in the manuscript in L294-299: "To validate the SOA model and to provide reference for calibration, we gathered experimental information on the four mentioned VOCs in section 2.1. The selected experimental studies were limited to those that provided all required variables: VOC concentration, reaction time, $NO_x$, ozone, RH, temperature, OH concentration, and final SOA mass. This was necessary because our modelling approach with SOMA uses these specific environmental and chemical parameters as direct inputs to simulate SOA formation. Without complete documentation of these conditions, the experiments would not be suitable for generating accurate model inputs or for evaluating model performance."

We acknowledge the reviewer's question on why GECKO-A was not directly run under the same experimental conditions. In the revised manuscript in L289-292 to explain: "The fitting curve allows us to calculate the SOA yield of any given time-period, for any $NO_x$ level. This interpolated fitting curve allows us to estimate SOA yield values at any $NO_x$ level without rerunning the GECKO-A model, which is necessary since we do not have access to the GECKO-A code but only to its published yield output library."

Regarding SOA loading effects, we focused on experiments without organic seed aerosol to avoid complications related to condensation sinks and OA background levels. In L304-307 we state: "Both studies were also selected because they did not include seed aerosol, in order to avoid uncertainties related to partitioning onto pre-existing OA. The data collected for every compound serve as reference for validation and as a base for experimental fitting based on the GECKO-A constant conditions to correct the SOA yields used by SOMA."

**Comment 4:**

4. Section 4.1: I do not agree with how the model was setup for background/boundary values of toluene. I don't see how a uniform background toluene assumption is justifiable given that the concentrations inside the modeled domain vary spatially. The modeled domain is identical to the regions surrounding it so if toluene varies inside the domain, that fact should also hold for regions immediately surrounding the domain.

**Response:**

We thank the reviewer for raising this important point. We agree that assuming a spatially uniform background can oversimplify real-world variability, especially in a heterogeneous urban environment. In response, we have revised Section 4.2.2 to clarify the rationale and limitations of our assumption. Specifically, we added the following text:

In L428-433: "Certain VOCs can remain in the air for hours to days, allowing them to recirculate in urban areas and contribute to SOA production. This contribution often originates from neighboring areas and can accumulate over various time scales. To estimate the contribution of SOA originating from neighboring areas to the study domain, we assume that traffic patterns are consistent throughout the broader urban region due to similar driving conditions of urban environments. This assumption was supported by the fact that the modelled domain lies centrally within Augsburg and is surrounded by areas with comparable land use, emission sources, and road networks."

L472-474 we also state: "We assumed that similar emissions and meteorological conditions in the broader urban setting lead to comparable VOC and SOA levels in the surrounding areas, justifying their use as boundary input."

These additions aim to make the reasoning behind our boundary setup transparent and acknowledge its potential impact on model results. We hope this adequately addresses the reviewer's concern.

**Comment 5:**

5. The strong suit of this work is the CFD modeling and what can be learned from it. The development of methods to estimate SOA mass yields are clunky at best. This work's Achilles heel is GECKO, which is great at gaining a fundamental understanding of multi-generational SOA chemistry and not a good fit for predicting SOA mass yields. Why not use published VBS parameterizations with NOx dependence to get at SOA mass yields directly? They may not be perfect but would work better than GECKO and allow the authors to focus on the CFD insights.

**Response:**
We appreciate the reviewer's point regarding the potential utility of VBS parameterizations. Our choice to use GECKO-A was driven by the desire for consistency across experimental conditions ($NO_x$, RH, T), and because it provides chemically-resolved, compound-specific SOA yields that serve as a transparent input to SOMA. To support our approach we explain in the introduction section in L61-69: "SOA formation models are commonly used to simulate secondary organic aerosol production. One widely used approach is the Volatility Basis Set (VBS) model, which simulates the partitioning of semi-volatile and intermediate-volatility organic compounds (S/IVOCs) between the gas and aerosol phases (Sasidharan et al., 2023). While VBS frameworks are robust and widely adopted for regional-scale chemical transport models, they typically require the definition of volatility bins and yield parameterizations that must be fitted to experimental data, and they do not include explicit chemical reaction pathways. In principle, VBS approaches can account for environmental dependencies such as $NO_x$ levels, temperature, or relative humidity by adjusting the bin yields, but this is often done indirectly and can require significant fitting effort. In contrast, our approach uses SOA yields generated by the GECKO-A model, which includes detailed multi-generational oxidation chemistry and accounts for precursor-specific reaction mechanisms."

To acknowledge the limitation of our approach but enhance our stand we state in L69-72: "We acknowledge that GECKO-A is not optimized for direct SOA mass prediction under atmospheric conditions; however, its mechanistic representation provides a chemically consistent foundation from which SOA yields can be generated."

We appreciate this valuable suggestion and will explore using VBS-based formulations in future iterations of our modeling approach.

Minor comments:

**Comment 7:**

Line 32: I haven't seen 'OHs' written in plural.

**Response:**

We corrected that inconsistency in the manuscript in L33-35: "OH react with VOCs, initiating chemical transformations that reduce VOC volatility and promote their condensation onto existing particles or formation of new ones (Hallquist et al., 2009)."

**Comment 8:**

Line 111: Provide references for the NOx levels used to generate GECKO output. Same goes for the choice of RH and T.

**Response:**

The GECKO-A output library's details are available online here: https://www2.acom.ucar.edu/modeling/gecko/details

In L130-135 we modified the manuscript to include a reference of the early development of GECKO: "The GECKO-A model produces SOA yields for five distinct NOx concentration levels, corresponding to remote (0.002 ppb), remote continental (0.0025 ppb), continental (5 ppb), polluted continental (20 ppb) and urban (200 ppb) environments (Camredon et al., 2007). Users can select a specific VOC and a $NO_x$ pollution scenario, and then visualize how SOA yield changes over time, with temporal resolution ranging from seconds to hours. The SOA yields given by GECKO-A correspond to constant conditions of temperature (T) at 25°C, ozone ($O_3$) concentration of 40 ppb and relative humidity RH at 70% (Camredon et al., 2007) ."

Citation: https://doi.org/10.5194/egusphere-2025-193-RC1

**References:**

Camredon, M., Aumont, B., Lee-Taylor, J., & Madronich, S. (2007). The SOA/VOC/NO x system: an explicit model of secondary organic aerosol formation. In *Atmos. Chem. Phys* (Vol. 7). www.atmos-chem-phys.net/7/5599/2007/

Chen, T., Zhang, P., Ma, Q., Chu, B., Liu, J., Ge, Y., & He, H. (2022). Smog Chamber Study on the Role of NO xin SOA and O3Formation from Aromatic Hydrocarbons. *Environmental Science and Technology*, *56*(19), 13654–13663. https://doi.org/10.1021/acs.est.2c04022

Deng, W., Liu, T., Zhang, Y., Situ, S., Hu, Q., He, Q., Zhang, Z., Lü, S., Bi, X., Wang, X., Boreave, A., George, C., Ding, X., & Wang, X. (2017). Secondary organic aerosol formation from photo-oxidation of toluene with NOx and SO2: Chamber simulation with purified air versus urban ambient air as matrix. *Atmospheric Environment*, *150*, 67–76. https://doi.org/10.1016/j.atmosenv.2016.11.047

Hallquist, M., Wenger, J. C., Baltensperger, U., Rudich, Y., Simpson, D., Claeys, M., Dommen, J., Donahue, N. M., George, C., Goldstein, A. H., Hamilton, J. F., Herrmann, H., Hoffmann, T., Iinuma, Y., Jang, M., Jenkin, M. E., Jimenez, J. L., Kiendler-Scharr, A.,

Maenhaut, W., … Wildt, J. (2009). The formation, properties and impact of secondary organic aerosol: current and emerging issues. In *Atmos. Chem. Phys* (Vol. 9). www.atmos-chem-phys.net/9/5155/2009/

Sasidharan, S., He, Y., Akherati, A., Li, Q., Li, W., Cocker, D., McDonald, B. C., Coggon, M. M., Seltzer, K. M., Pye, H. O. T., Pierce, J. R., & Jathar, S. H. (2023). Secondary Organic Aerosol Formation from Volatile Chemical Product Emissions: Model Parameters and Contributions to Anthropogenic Aerosol. *Environmental Science and Technology*, *57*(32), 11891–11902. https://doi.org/10.1021/acs.est.3c00683

---

## Author Comment (AC2)

**Response to comments on "Development and implementation of SOMA: A Secondary Organic Module for Aerosol integration in high-resolution air quality simulations"**

We would like to thank the reviewers for evaluating our manuscript and for providing their constructive comments. Below you can find point-by-point answers to the RC2 reviewer's comments. All references are provided at the end.

**RC2: 'Comment on egusphere-2025-193', Anonymous Referee #2, 31 Jul 2025**

This paper presents a novel model named SOMA, which represents Secondary Organic Aerosol (SOA) formation in a way that the authors claim is suitable for use in complex atmospheric chemistry models and capable of including correction factors depending on various inputs which can influence SOA formation including ozone levels, relative humidity, and temperature. The authors have so far created a version of this model capable of representing four key SOA-forming compounds based on laboratory experiments: alpha-pinene, isoprene, limonene, and toluene. The authors then apply this model in a case study to characterize toluene SOA production from traffic sources in Augsburg Germany, using a recirculating model framework (essentially a periodic boundary condition) which enables them to represent their system and boundary conditions with a single model. Using this framework, the authors are able to conclude (for a simplified system of toluene aerosols) that background sources contribute 21-53% of SOA over the course of the day, and that the influence of any given background signal degrades to nothing over the course of 7 hours due to precursor consumption and dilution.

The paper advances two interesting ideas, and deserves consideration. However, it also needs significant revisions of both text and ideas before it should be published.

**Main Points**

**MAIN POINT 1:**

**SOMA vs VBS**

The authors make a strong claim that a main advantage of their model vis a vis a more traditional volatility basis set (VBS) framework lies in the ability of SOMA to simulate complex chemistry, while the main advantage SOMA has over more complex semi-explicit chemical mechanism models lies in its reduced computational costs.

I do not entirely agree with the authors' framing here, specifically regarding the comparison of SOMA and VBS-type techniques, and in fact I would foreground an entirely different strength of their model. It does not seem to me that there is any process represented in SOMA that could not be represented with a sufficiently complex VBS framework. The authors highlight the role of ozone levels, relative humidity, and temperature, but all of these could, in principle, be represented in VBS frameworks as they already exist. In my view, the main advantages of this model over the VBS framework are:

- the simplicity of the calculations

1. Where a VBS might require tracers representing 3-6 volatility bins per SOA precursor, SOMA applies parameterized gamma and lambda terms to calculate an overall yield. As the authors move beyond 4 SOA precursors, the enormous advantages of this construction over the VBS framework will become even more obvious.

- the transparency of the assumptions and

- the interpretability of the intermediate steps.

   1. SOMA generates an SOA yield term! This is immediately comprehensible to any experimentalist and can be readily understood by any reader or other modeler in a manner that the complex partitioning theory of a VBS makes challenging.

   2. Since this is how measurements of SOA formation are taken *in a lab* in any case, comparing model input with new experimental data does not require a complex process of fitting volatility bin yields to experimental observations in order to back out the desired OA yield.

These are not minor advantages, and I encourage the authors to highlight them!

Additionally, SOMA has one drawback when compared to a VBS framework: it assumes implicitly that the partitioning of low-volatility gases into the particle phase is unidirectional and irreversible. The authors do not discuss this assumption, and they may wish to do so.

**Response:**
We sincerely thank the reviewer for this thoughtful and constructive comment. We fully agree that the strengths of SOMA should be more clearly stated, especially in comparison to VBS-type frameworks. Based on the reviewer's suggestions, we have revised the relevant parts of the manuscript to better highlight SOMA's main advantages in L63-72: "While VBS frameworks are robust and widely adopted for regional-scale chemical transport models, they typically require the definition of volatility bins and yield parameterizations that must be fitted to experimental data, and they do not include explicit chemical reaction pathways. In principle, VBS approaches can account for environmental dependencies such as $NO_x$ levels, temperature, or relative humidity by adjusting the bin yields, but this is often done indirectly and can require significant fitting effort. In contrast, our approach uses SOA yields generated by the GECKO-A model, which includes detailed multi-generational oxidation chemistry and accounts for precursor-specific reaction mechanisms. We acknowledge that GECKO-A is not optimized for predicting absolute SOA mass yields under all atmospheric conditions; however, its mechanistic representation provides a chemically consistent and precursor-resolved foundation from which SOA yield trends can be generated".

We agree that we should highlight the transparency in our method and assumptions. We refer to that in L78-81: "Additionally, correction factors based on experimental SOA formation data

from literature are used to account for the effects of other influential factors, such as relative humidity (RH), and temperature (T) as well as other oxidants like ozone ($O_3$). This correction step is applied directly to yield values using regression-based adjustment factors, providing a transparent and intuitive method to account for environmental variation."

Regarding the assumption of irreversible partitioning, we agree that this is a simplification, and we have now explicitly acknowledged this assumption L83-84: "We also note that SOMA assumes irreversible partitioning of semi-volatile products into the particle phase, which is a simplification commonly used in mechanistic studies when simulating fresh SOA formation over short time scales."

We thank the reviewer again for their insightful reframing of our model's strengths, which has significantly improved our presentation and justification of SOMA.

**MAIN POINT 2:**

**The Recirculation Approach**

The authors repeatedly stress the 'innovation' and 'novelty' of their recirculation modeling approach (though the distinctions between it and the cyclic/periodic boundary condition options available in such models as OpenFOAM elude me somewhat), but they do not sufficiently expand on *why* the approach might be specifically valuable, nor what might make it applicable elsewhere. Section 4.2.2 dispenses with this explanation in a single phrase: "traffic-related emissions can be considered representative of the surrounding zones, given the study area's central location within the city of Augsburg". They do not expand further on why this approach might be advantageous. This atmospheric modeler recognizes that the recirculation approach that the authors have used here greatly simplifies the treatment of what would otherwise have to be independently-generated boundary condition inputs. In fact, this fine point of modeling concept is key to understanding the authors' designation of F/BG/BGF-SOA in section 4.2.2 (see my confusion regarding the discussion from lines 392-403), but is not spelled out in the paper.

Additionally, given that the authors identify a 7-hour window over which the background toluene SOA formation might contribute to observed concentrations, there might be a useful exercise regarding how large a city airshed would have to be for this recirculation approach to be representative, or alternately a quantification of how much recirculation is present in the true atmosphere over a city in question, in order to determine whether the recirculation model might be appropriate for use.

**Response:**
We thank the reviewer for this insightful comment, which helped us realize the need to better articulate both the purpose and the advantages of the recirculation approach introduced in this study. To clarify these points, we have significantly revised Section 4.2.2 to provide a clearer justification and explanation. Specifically, we have included the following text in L428-435:

"Certain VOCs can remain in the air for hours to days, allowing them to recirculate in urban areas and contribute to SOA production. This contribution often originates from neighboring areas and can accumulate over various time scales. To estimate the contribution of SOA originating from neighboring areas to the study domain, we assume that traffic patterns are consistent throughout the broader urban region due to similar driving conditions of urban environments. This assumption was supported by the fact that the modelled domain lies centrally within Augsburg and is surrounded by areas with comparable land use, emission sources, and road networks. By focusing on our specific case-study area, we argue that traffic-related emissions can be considered representative of the surrounding zones, given the study area's central location within the city of Augsburg (Figure 2a)."

To clarify the designation of F-, BF-, BGF-SOA we explain more comprehensively in L439-451: "To clarify, SOA in the modelling domain can have three different origins:

1. Freshly formed SOA resulting from VOC emissions produced within the domain during each hour modelled. This is referred to as fresh SOA **(F-SOA)**.
2. Previously formed SOA outside of the modelling domain, transported into the study area within the given hourly time frame, referred to as background SOA **(BG-SOA)**.
3. Freshly formed SOA resulting from VOC transported into the modelling domain from neighboring areas, referred to as background fresh SOA **(BGF-SOA)**.

We group BG-SOA and BGF-SOA together as "background SOA," representing all non-local contributions to total SOA during each hourly period. This recirculation approach allows us to estimate background and delayed SOA contributions using only local emission and meteorological data, without requiring external boundary condition inputs. This enables clear differentiation between fresh and recirculated SOA mass, as defined by the F-SOA, BG-SOA, and BGF-SOA categories. Equation 11 is used to calculate the total SOA mass concentration.

$$SOA_{total}=(F\text{-}SOA)+(BG\text{-}SOA)+(BGF\text{-}SOA) \tag{11}"$$

We thank the reviewer for giving us the opportunity to also clarify and better explain the purpose and advantages of our recirculation approach. In L526-532 we explain: "This stabilization occurs because the volatile organic compounds (VOCs) emitted in earlier time periods have either been consumed through chemical reactions or broadly diluted in the atmosphere. As a result, after 7 hours, the cumulative effect of background SOA becomes negligible. This 7-hour threshold provides a useful estimate of the effective atmospheric memory of traffic-related SOA within the modelling domain. It suggests that, under similar urban conditions, the recirculation-based background contribution is most relevant within this timeframe and declines afterwards. Therefore, the recirculation approach used here may be applicable to other urban-scale studies, provided the city's emission dynamics and atmospheric conditions support a comparable chemical lifetime and dispersion behavior."

***Specific Comments***

**Comment 1:**

Line 31-32: citing 'Saiz-Lopez et al., 2017' for the statement that 'hydroxyl radicals have a great influence on SOA production as they are the most potent oxidant in the atmosphere' seems like an odd choice. This is neither a new nor particularly controversial statement – why include this citation? If the authors *do* want to include a cite here, Calvert et al. (2002) might be an appropriate citation, at least for aromatic hydrocarbons such as toluene.

Calvert, J. G., Atkinson, R., Becker, K. H., Kamens, R. M., Seinfeld, J. H., Wallington, T. J., and Yarwood, G.: The Mechanisms of Atmospheric Oxidation of Aromatic Hydrocarbons, Oxford University Press, New York, 556pp., 2002.

**Response:**
We thank the reviewer for this suggestion and agree. We have replaced the (Saiz-Lopez et al., 2017) citation with the more foundational (Calvert et al., 2002) reference, which better reflects the established understanding of OH reactivity with aromatics like toluene. In L31-32 in the revised document we state: "Hydroxyl radicals (OH) have a great influence on SOA production as they are the most potent oxidant in the atmosphere (Calvert et al., 2002)."

**Comment 2:**

Lines 44-45: citing two papers from 2014 and 2017 for the source of 'oxidation chambers' seems like an odd choice, especially since the paper also cites earlier work on the same topic (e.g, both of the Ng. et al 2007 papers cited). The authors should re-examine these citations and consider more foundational choices, unless there is a specific reason why these two studies are being cited. Similarly for the OFR reactions, the original citations should probably be those of Kang et al (2007) and Lambe et al. (2011)

Kang, E.; Root, M. J.; Toohey, D. W.; Brune, W. H. Introducing the Concept of Potential Aerosol Mass (PAM) Atmos. Chem. Phys. 2007, 7, 5727– 574417

Lambe, A. T.; Ahern, A. T.; Williams, L. R.; Slowik, J. G.; Wong, J. P. S.; Abbatt, J. P. D.; Brune, W. H.; Ng, N. L.; Wright, J. P.; Croasdale, D. R. Characterization of Aerosol Photooxidation Flow Reactors: Heterogeneous Oxidation, Secondary Organic Aerosol Formation and Cloud Condensation Nuclei Activity Measurements Atmos. Meas. Tech. 2011, 4, 445– 46118

**Response:**
We thank the reviewer for this comment, we agree that we should include more fundamental work to cite oxidation chamber and OFR studies. In L43-45 in the revised manuscript we state: "To quantify the amount of SOAs formed under controlled conditions, researchers utilize specially designed oxidation chambers (Surratt et al., 2006) and oxidation flow reactors (OFRs) (Kang et al., 2007; Lambe et al., 2011)."

**Comment 3:**

Line 63: 'the VBS approach does not include detailed chemical reactions' – I do not entirely agree with this statement, and I think the authors should refine the claim they are trying to make here. The VBS necessitates simplification of chemistry *on the basis of volatility*, and I think this is what the authors mean? But this does not precisely preclude the inclusion of detailed

chemical reactions, it simply abstracts them behind the layer of partitioning theory and volatility bins.

**Response:**

We acknowledge the imprecise wording and in L63-69 we have revised the statement to: "While VBS frameworks are robust and widely adopted for regional-scale chemical transport models, they typically require the definition of volatility bins and yield parameterizations that must be fitted to experimental data, and they do not include explicit chemical reaction pathways. In principle, VBS approaches can account for environmental dependencies such as $NO_x$ levels, temperature, or relative humidity by adjusting the bin yields, but this is often done indirectly and can require significant fitting effort. In contrast, our approach uses SOA yields generated by the GECKO-A model, which includes detailed multi-generational oxidation chemistry and accounts for precursor-specific reaction mechanisms."

**Comment 4:**

Line 68: This is why the prior statement matters. This reviewer is familiar with VBS frameworks capable of considering NOx levels, SOA yields, oxidation duration, and OH concentrations. Doing all of these things in a single VBS comes with significant drawbacks to computational efficiency, as it necessitates a large number of reactions representing different yields of individual volatility bins and additional reactions passing mass from one volatility bin to another.

**Response:**

We thank the reviewer for the constructive and helpful comment. We now elaborate more in the manuscript regarding the positives of our approach in L76-82: "This model incorporates key variables such as NOx levels, SOA yields, oxidation duration, and hydroxyl radical (OH) concentrations specific to the environment being studied. The GECKO-A model is used for the calculation of the SOA yields used. Additionally, correction factors based on experimental SOA formation data from literature are used to account for the effects of other influential factors, such as relative humidity (RH), and temperature (T) as well as other oxidants like ozone ($O_3$). This correction step is applied directly to yield values using regression-based adjustment factors, providing a transparent and intuitive method to account for environmental variation. The model is calibrated using experimental data allowing for greater accuracy while maintaining computational efficiency."

**Comment 5:**

Line 80: "No study has yet modelled and quantified the contribution to local SOA mass over various time scales in urban environments, highlighting a gap in current research" is a bold claim to make. The authors should reconsider what precisely they mean to say here, as I do not think that the statement is accurate as written.

**Response:**

In L97-99 we revised this statement to be more precise and defensible: "To our knowledge, no study has quantified the contribution of background and local SOA at high spatiotemporal resolution in urban environments using CFD-based dispersion modelling."

**Comment 6:**

Figure 1 and discussion: how should readers interpret these GECKO-corrected yields in the context of their use in SOMA, given that they change over time while SOMA gives a single yield term, albiet corrected for temperature, ozone, and RH? Also, is there a logic behind the color choices here?

**Response:**

We thank the reviewer for pointing this out. Figure 1 provides GECKO-A yields over time, before the experimental correction. In L139-140 we explain: "Figure 1 shows trends in SOA yields from toluene ($C_7H_8$), limonene ($C_{10}H_{16}$), isoprene ($C_5H_8$) and α-pinene ($C_{10}H_{16}$) under urban conditions ($NO_x$ = 200 ppb) generated by GECKO-A, as examples." The colours are just used to distinguish between the compounds.

[Figure]

Figure 1. Time series of SOA yields from GECKO-A for urban conditions ($NO_x$=200 ppb), for toluene (a), limonene (b), isoprene (c) and α-pinene (d) for an initial organic mass loading of 1 pptvC.

**Comment 7:**

Line 123-124: Ng et al. 2007 citation here is getting treated oddly (*Ng, Kroll et al. 2007*) since it is one of two Ng et al. 2007 cites in this paper. As someone who has fallen victim to this exact problem with precisely these two papers, I have sympathy. I think the preferred style is (Ng et al. 2007a; Ng et al. 2007b), but the authors may want to double check.

**Response:**

We appreciate the reviewer for pointing this out. We have now corrected the text to distinguish between both citations.

In L54-55 we state: "Several factors influence the formation of SOAs. Studies have shown a clear trend of increasing SOA yield with decreasing NOx concentrations (Brégonzio-Rozier, 2015; Ng et al., 2007a)."

In L142-144 we edited the text accordingly: "For toluene (Figure 1a), SOA yield exhibits a sharp initial increase, stabilizing at around 0.6%, suggesting a balance between formation and loss processes influenced by NOx levels (Ng et al., 2007b)."

**Comment 8:**

Figure 4: I appreciate this figure! It's a helpful mental map. One question: why is limonene shaded a completely different color from the other compounds?

**Response:**
Thank you for pointing this out. The colour has been adjusted for consistency with the other VOCs. There was no scientific reason for the distinction.

[Figure]

Figure 4. SOMA uses [VOC], [NOx], [OH], time and $\Delta t$ as an input to calculate the amount of SOA generated. SOA yields are selected from the GECKO-A output library. The model uses correction factors accounting for the influence of $O_3$, T and RH based on experiments

**Comment 9:**

Line 285/Section 3.4: Does this model contemplate the oxidation of species which are primarily oxidized by $O_3$ rather than OH? It's possible this question is outside of the scope of this paper, but it would become relevant if models like this one became more widely used.

**Response:**
Thank you for this constructive comment. SOMA in this primary version demonstrated in this paper contemplates the influence of $O_3$ oxidation by using experimental data. The equation used to calculate the amount of SOA formed only takes into account OH presence. Equation (9) in L269:

"$SOA = [VOC] \cdot (1 - e^{-kOH \cdot [OH] \cdot \Delta t}) \cdot Y \cdot (\lambda_{cor})$                    (9)"

In the same equation the $\lambda_{cor}$ factor is created for each compound to contemplate the influence of other parameters such as the oxidation by $O_3$. This is how SOMA incorporates $O_3$ oxidation in its current version. For clarification we state in the manuscript in L262-267: "This approach does not account for the influence of temperature and relative humidity, factors that affect the formation of SOA and other oxidants like ozone. To address this limitation and improve the model's reliability, we introduce a correction factor ($\lambda_{cor}$) into the equation (Equation 9). This factor is produced by collecting experimental information, as shown in Figure 4, for every investigated compound to determine the influence of other parameters ($O_3$, T, and RH), given that the yields taken from GECKO-A assume constant conditions with $O_{3gecko}$= 40 ppb, $T_{gecko}$= 25°C and $RH_{gecko}$=70% as explained in section 2.1."

**Comment 10:**

Lines 300-320: it is not at all clear to this reader \*how\* the authors managed to account for temperature dependence based on their source experiments. As both the text and Figure 6 make clear, the ratio of $T_{exp}/T_{gecko}$ was essentially 1 for all cases. Despite this, the authors claim to be able to account for temperature dependence both in their introduction/abstract and explicitly in Equation (10). How this was achieved is unclear. Is the answer just: 'in principle SOMA could do this assuming the relevant experiments had been performed, but in the case of toluene no such experiments exist'? If so, isn't the inclusion of a temperature correction at all potentially misleading? What happens if you use this temperature relationship (derived across the range of temperatures from 25.8-26.9 C) at some atmospherically relevant temperature outside of the range of the parameterization?

**Response:**
We agree that the influence of temperature is not great in this case. We explain in the manuscript in L330-335: "The $T_{exp}/T_{gecko}$ ratios in all cases are very close to 1, as the temperatures examined in the experiments ranged from 25.8 to 26.9 °C, and the GECKO-A temperature is considered constant at 25 °C. As such, the influence of temperature is minimal in the toluene-specific dataset and any fitted temperature dependence here should be interpreted with caution. We include the temperature correction term in Equation 10 primarily to retain consistency across compounds, as in other VOC cases (SI) experimental temperatures vary more significantly and show a clearer influence on SOA yield. Thus, the regression method is designed to support multi-compound calibration and not solely optimized for toluene."

**Comment 11:**

Figure 7: This figure is compelling but could be considerably clearer. For one thing, the labels in the caption and on the figure do not correspond to one another. The authors should make the fact that 'original model' (I believe this is uncorrected GECKO-A output, currently in orange) and 'corrected' (I believe this to be the SOMA output, currently in green) are model outputs, and that purple is the experimental output and \*NOT\* from any model. This could easily be achieved with 1) a better caption and 2) a clear connection between the two modeled outputs – perhaps as shades of the same color, or the same color with different textures applied. Additionally, the y-axis label is unnecessarily unclear.

We thank the reviewer for addressing this. We revised the caption to explicitly state:

- Orange = original SOMA using GECKO-A yields

- Purple = Experimental data

- Green = corrected SOMA outputs

The y-axis refers to the exact concentrations measured at the experiments and the calculated original and corrected SOMA outputs, using the initial conditions from each experiment in Figure 7 in L355:

[Figure]

Figure 7. Comparison between experimental SOA concentrations, original GECKO-A estimates and SOMA predictions in the case of toluene. TOL1-9 represent the experiments of SOA formation from toluene from Table 3.

**Comment 12:**

Figure 8: I think this figure could be shifted to the supplemental materials – it is unclear to me that it is much more relevant than the GECKO-A output in Figure 5 as it only summarizes the effect of the inputs already in GECKO-A, rather than the effects of the new factors which SOMA can account for (RH, T, O3). If it is to be kept in the paper, the Y-Axes should be relabeled to be in percent, as the current y-axis is opaque. Also, the label on the figure 'SOA toluene' does not match with the caption 'toluene-to-SOA' formation. Neither label makes much sense, and both should be re-thought.

**Response:**
We thank the reviewer for the ideas and suggestions. We believe that Figure 8 should remain in the manuscript because it shows the sensitivity of SOMA using corrected GECKO-A yields in terms of OH and NOx presence in calculating SOA formation as stated in L358-366: "After calibration, SOMA can be used to explore the impact of different parameters on SOA

formation. Figure 8 shows the influence of OH and $NO_x$ on SOA formation within 1 h exposure of 1 µg/m³ for toluene in typical atmospheric conditions. Figure 8a demonstrates that within the range of specific OH levels ($10^5$ to $10^7$ molecules/cm³), the toluene SOA mass increases with increasing OH concentration (Sarrafzadeh et al., 2016). Beyond $5 \times 10^6$ OH molecules/cm³, the SOA mass reaches a plateau, indicating a maximum yield. Once OH radicals have oxidized all the precursors contributing to SOA formation, the remaining unoxidized precursors can no longer participate in SOA mass growth, because of the insufficient availability of oxidizing agents (Song et al., 2019). In Figure 8b, $NO_x$ concentration exhibits an inverse relationship with SOA formation, aligning with the trends observed in the yield selection process from GECKO-A based on compound, $NO_x$ concentration and time described in section 3.2 (Figure 5)."

We agree that the y-axis should change to percentage to make the effect more understandable. We also corrected the caption of Figure 8 in L370 to be consistent with the SOA toluene label: "

[Figure]

Figure 8. Influence of (a) OH and (b) $NO_x$ concentrations on SOA toluene formation, demonstrated by the SOMA model, by using initial concentration of 1 µg/m$^3$ of toluene for 1hr for every case."

**Comment 13:**

Line 349: delete the 'and' in this line to make the claim clearer: 'toluene is …the only traffic-linked SOA precursor for which … measurements were available'.

**Response:**
in L383-385 we deleted the and to make the claim clearer: "Toluene was chosen because it is the only SOA precursor that can be directly linked to traffic in a city environment for which actual concentration measurements were available."

**Comment 14:**

Line 353: how good is this assumption of a uniform background concentration within the modelled domain given the atmospheric lifetime of toluene?

**Response:**
Thank you for this comment. In L387-397 we elaborate further to explain our rationale: "CFD

concentration predictions were incrementally added to toluene concentrations measured at an air quality station outside of the city (LFU station - Figure 2a), that served as the background. This approach assumes a uniform background concentration within the modelled domain to account for the impact of sources not related to urban emissions (Tchepel et al., 2010). Given the moderate atmospheric lifetime of toluene and the limited spatial extent of the domain, this assumption is considered reasonable over the modelled time period. The LFU background station is located away from direct traffic influence and provides stable background values that we superimpose with CFD-simulated traffic concentrations to reconstruct total levels at the KP station."

**Comment 15:**

Lines 363-365: These sentences seem backwards to me, as I understand them. The authors identified the toluene attributable to traffic activity \*BY\* subtracting out the background toluene, which left them with a toluene concentration they could compare to model output. Is that correct? I'm left slightly uncertain, because as written this is confusing.

**Response:**
Thank you for pointing this out. In this section we actually performed two kinds of comparisons for validation. In L402-406 we make our statement clearer by elaborating: "In this section we also performed direct comparison of CFD generated toluene concentrations that are attributed to only traffic activity. To do that we identified the part of the measured toluene concentration that could be attributed to traffic activity. By subtracting the corresponding toluene concentrations measured at the background station (LFU) located on the southern side of the urban area for the same hour from the measurement at KP, we can determine the street increment of the measurement."

**Comment 16:**

I think my prior comment relates to my confusion with Figure 9: both the design of this figure and its discussion should make clearer what, precisely, is being compared to what here. As I understand it, the black line meant to show total measurement, and the green line is the measurement at KP-LFU. So far so good. But now, what is being represented in the orange and green bars? These are model outputs with background toluene and the traffic-toluene represented. So, should \*only\* the green bars be compared with the green line? Should the Green+Orange be compared with the black line? The conclusion reached: " The agreement with measured toluene concentrations suggests that the model effectively replicates the dispersion of traffic-related toluene within the urban environment" suggests that the comparison ought instead to be between the green line and green bars. But if this is the case, why even show the orange bars in the figure? The more I read this section, the more confused I get.

**Response:**
We understand the confusion that may have been cause from our poor wording. To make our statements clearer in L393-395 we elaborate: "Figure 9 demonstrates good agreement between measured (black curve) and estimated toluene concentrations (orange and striped green bars)

over a 24-hour period, with an average deviation of 8% and a correlation coefficient of 0.83."
To explain the comparison between only the CFD outputs and the street increment of the toluene measurements we state in L411-414: "While the hourly comparison showed some negative increments—due to instances when the background measurement was higher than the KP station measurement (between 11−12 am and 1−2 pm, as shown in Figure 9)—the daily concentrations obtained from both the CFD model, and the street increment, meaning comparing the green curve and the stripped green bars, showed a deviation of 15%."

**Comment 17:**

Line 383: The authors should include a very brief discussion of how their choice of OH concentration compares to other urban environmental studies or observations. I know how that [OH] lines up but the average reader may or may not.

**Response:**
We thank the reviewer for pointing this out. To clarify our selection we discuss further in L420-425:
"In urban environments, the OH levels vary between $0.7 – 8\times10^6$ molecules/cm$^3$ depending on light intensity, temperature and the presence of certain pollutants (Cho et al., 2023). No information on urban OH levels was available for this study, so we selected an OH concentration of $2\times10^6$ molecules/cm$^3$. This value falls within the lower-middle range of typical OH concentrations reported in urban studies (Zou et al., 2023) and reflects representative daytime conditions in polluted environments. We used a conservative estimate to avoid overestimating SOA formation, given the absence of direct OH measurements in Augsburg."

**Comment 18:**

Lines 392-400: the paragraph and the numbered list are repetitive and could be condensed down to just the list with a few additions. Additionally, couldn't we also consider a fourth category of 'Freshly formed SOA resulting from VOC emissions produced within the domain during an hour prior to the hour modeled'? From the discussion further down on page 20, it appears that this gets added into BGF-SOA – is that correct? On re-reading the paper, I think this choice is because of the recirculation assumptions, but the authors might considering making that explicit.

**Response:**
We thank the reviewer for this suggestion. The proposed fourth category "freshly formed SOA resulting from VOC emissions produced within the domain during the hour prior to the one modelled" is conceptually sound and is already included in what we define as background fresh SOA (BGF-SOA).

In L444-445:

1. "Freshly formed SOA resulting from VOC transported into the modelling domain from neighboring areas, referred to as background fresh SOA **(BGF-SOA)**."

**Comment 19:**

Line 405: something is odd with the formatting of equation (11). Specifically, the dash (e.g., in F-SOA) has been converted to an em-dash, which makes it look like a minus sign. This should be edited for clarity.

**Response:**
We thank the reviewer for pointing this out. We have now corrected the format of equation (11) in L451:
"$SOA_{total}$=(F-SOA)+(BG-SOA)+(BGF-SOA)                                        (11)"

**Comment 20:**

Line 419-421: Equations 12-14 are also oddly formatted and should be edited. If necessary, the authors should work with journal editors to get the effect they want. The authors currently appear to want to use two separate subscripts (e.g. VOC with subscripts [domain] and [t-1]). The result is an equation that is difficult to parse and confusing to look at. In equation (14), for example, there are two different lengths of 'minus' signs, which are very confusing. The same issues continue to plague the paper in equations 15-16.

**Response:**
We thank the reviewer for this observation. We agree that the equations were not written clearly to indicate which concentration is used and at which time period in the equations. To highlight the time-period selected we used [] for clarity in the equations and in the remaining text. We also corrected for any inconsistencies referring to hyphen and minus signs. In L465-467 we edited equations 12-14:

"$SOA_{domain}[t-1]= fSOMA(VOC_{domain}[t-1])$                                        (12)

$\Delta VOC_{domain}[t-1]=[VOC_{domain}[t-1]\cdot(1-e^{-kOH\cdot[OH]\cdot\Delta t})$                    (13)

$VOC_{remaining-domain}[t-1]=VOC_{domain}[t-1]-\Delta VOC_{domain}[t-1]$                (14)"

In L468-469 we changed the t-1 to [t-1] in every case:
"The terms $SOA_{domain}[t-1]$ and $VOC_{remaining-domain}[t-1]$ represent the average SOA formed and remaining VOC in the domain respectively as calculated by equations 12 and 14 for time [t-1]."

In L476 equation 15 is edited:
"$BG-SOA_t= fCFD_{background}(SOA_{domain}[t])$                                        (15)"

In L480 equation 16 is edited:
"$VOC_t=fCFD_{background}(VOC_{remaining-domain}[t])$                                    (16)"

**Comment 21:**

Line 446: once again, the authors have chosen some odd citations (why cite Wang et al. 2021?) for the obvious point that 'photochemistry occurs during periods of sunlight'. The entire sentence beginning with 'SOA is mostly formed during periods with UV radiation' is unnecessary given the prior sentence and the discussion of this idea on lines 32-33 of the paper.

**Response:**

We thank the reviewer for this comment. We removed the sentence in section 4.3. However we added it in the introduction section to highlight that SOA is formed mainly during sunlight, in the presence of OH radicals, to justify our selected period for modelling (between 06:00-18:00). In L32-33 of the revised manuscript we added: "SOA is mostly formed during periods with UV radiation to facilitate the formation of hydroxyl radicals (OH) via sunlight exposure (Altshuller, 1989)."

**Comment 22:**

Figure 11: I think this is an effective figure, and it is sufficient to make the point. However, because there are so many colors, it can be challenging to intercompare between pies to figure out 'how long does it take for one of these times to stop mattering. If the authors wanted to rethink this plot, they could use line-and-area plots instead, so that you can follow a single hour along the x-axis until it disappears. However, this might lose the 'local' vs 'background' distinction, so if that is what the authors prefer to highlight, the figure is ok as-is.

Additionally, some house-keeping on this figure: If the yellow pie slice in the '14:00-15:00' plot is striped, I cannot see it on my copy of the paper, so this color may need to be adjusted. Also, I think that in the caption, the authors meant to write 'striped slices' rather than 'stripped slices'.

**Response:**

We thank the reviewer for the positive feedback on Figure 11 and for the helpful suggestions to improve its clarity. To clarify our selection in using striped and solid-colour slices in L503-504 we report: "To clearly distinguish local from background SOA contributions, striped slices are used to indicate locally formed SOA, while solid-colour slices represent background SOA contributions."

We acknowledge that the yellow pie slice in the '14:00–15:00' period of Figure 11 is not clearly visible, and in L510 we adjusted its colour to improve visibility and corrected the

caption from stripped to striped:

[Figure]

Figure 11. Local (striped slices) and background contribution on SOA concentration from traffic-toluene emissions produced by recirculation approach."

Regarding the reviewer's insightful suggestion to use line-and-area plots to better visualize the decay or disappearance of contributions over time, we would like to note that Figure 12 in the manuscript serves a similar purpose. In L512-517 we introduce the concept and purpose of Figure 12: "This analysis allows us to assess the cumulative contribution of SOA formed over different timescales, helping to identify when older SOA formations have a negligible impact on current SOA concentrations. In Figure 12, the SOA concentration at time=0 represents locally formed, fresh SOA (F-SOA) for each period examined. For any given period, the total SOA originating from time=0 corresponds solely to the freshly formed SOA. As time progresses, the total SOA in each period increases as SOA from prior time intervals is incorporated, adding background-originated SOA mass to the overall modelled SOA."

**Comment 23:**

Figure 12: I think this figure would be more intuitive to read if x-axis were reversed, with the 0 were on the right. Additionally, the y-axis label and figure caption should be harmonized rather than using different terminology between the two, and the x-axis label should be more

descriptive. If the x-axis is reversed, it might be easy for the x-label to read 'hours before modeled time period'.

**Response:**

We thank the reviewer for the suggestion. We reversed the x-axis of Figure 12 to be easier for the reader to understand our point and the title and the y-axis label to be consistent in L535:

[Figure]

Figure 12. Cumulative SOA concentration from traffic-toluene at the sensor location. The x-axis indicates how many hours ago the SOAs were formed and have an influence on SOA concentrations in every time-period.

**Comment 24:**

Line 490: 'making these conclusions broadly representative' but you just said they weren't representative? Perhaps the authors mean 'relevant' here rather than 'representative'?

**Response:**

We thank the reviewer for this comment. We have now changed our statement in the revised manuscript as follows in L542-544: "Nevertheless, toluene is a significant component of traffic emissions, making these conclusions broadly relevant for understanding traffic-related SOA formation in urban environments."

**Comment 25:**

Lines 525-526: The authors close with a forceful statement "Since SOAs cannot be directly linked to specific sources and are not currently regulated, this study is crucial for uncovering their formation processes and advancing knowledge in this area". This statement, while correct in its general gist, contains two strong claims, neither of which I agree with entirely. The authors may want to reconsider what exactly they mean here. SOA cannot *always* be linked to specific sources, but the use of Aerosol Mass Spectrometers allows for a certain amount of source fingerprinting (e.g., pyrogenic vs biogenic vs cooking vs hydrocarbon-like organic

aerosols), so it's an overstatement to say categorically that SOA 'cannot be directly linked to specific sources'. Additionally, are SOA not currently regulated at all? Air quality regulations often target a reduction in PM (of which SOA is a component) and control of specific VOC emissions (many of which are SOA precursors). I think it is correct to say that regulating SOA is difficult because the system is so complex and poorly understood. For this precise reason, identifying the precise effects of a known SOA precursor such as toluene is valuable! Similarly, identifying the effects of 'natural' SOA formation gives policymakers valuable information in terms of understanding what 'background' SOA concentrations would look like. All of this is doubly true if different types of SOA have different health effects, which is certainly a possibility we as a field have not yet ruled out.

**Response:**

We thank the reviewer for this insightful comment and for pointing out the need to clarify and refine our concluding statements. To do that in L578-586 we elaborate on the impact of our study and the need for continuous research on SOA formation: "While SOA cannot always be directly linked to specific sources, advances such as Aerosol Mass Spectrometry (AMS) have enabled partial source attribution based on composition profiles, distinguishing between biogenic, anthropogenic, or combustion-related contributions. Similarly, although SOAs are not individually regulated, their precursors and total particulate matter (PM) are typically targeted by air quality regulations. This study contributes to the effort of improving mechanistic understanding of SOA formation, particularly from key anthropogenic precursors like toluene, and can support the refinement of regulatory strategies by improving attribution and prediction of SOA impacts. Moreover, considering the potential variability in health impacts from different SOA types, identifying precursor-specific contributions is increasingly important for public health and environmental policy."

**Citation**: https://doi.org/10.5194/egusphere-2025-193-RC2

**References:**

Altshuller, A. P. (1989). Ambient Air Hydroxyl Radical Concentrations: Measurements And Model Predictions. *Journal of the Air Pollution Control Association*, *39*(5), 704–708. https://doi.org/10.1080/08940630.1989.10466556

Brégonzio-Rozier, L. (2015). Supplement of Gaseous products and secondary organic aerosol formation during long term oxidation of isoprene and methacrolein. *Supplement of Atmos. Chem. Phys*, *15*, 2953–2968. https://doi.org/10.5194/acp-15-2953-2015-supplement

Calvert, J. G., Atkinson, R., Becker, K. H., Kamens, R. M., Seinfeld, J. H., Wallington, T. J., & Yarwood, G. (2002). *The Mechanisms Of Atmospheric Oxidation Of Aromatic Hydrocarbons*. Oxford University Press. https://doi.org/10.1093/oso/9780195146288.001.0001

Cho, C., Fuchs, H., Hofzumahaus, A., Holland, F., Bloss, W. J., Bohn, B., Dorn, H. P., Glowania, M., Hohaus, T., Liu, L., Monks, P. S., Niether, D., Rohrer, F., Sommariva, R.,

Tan, Z., Tillmann, R., Kiendler-Scharr, A., Wahner, A., & Novelli, A. (2023). Experimental chemical budgets of OH, HO2, and RO2 radicals in rural air in western Germany during the JULIAC campaign 2019. *Atmospheric Chemistry and Physics*, *23*(3), 2003–2033. https://doi.org/10.5194/acp-23-2003-2023

Kang, E., Root, M. J., Toohey, D. W., & Brune, W. H. (2007). Atmospheric Chemistry and Physics Introducing the concept of Potential Aerosol Mass (PAM). In *Atmos. Chem. Phys* (Vol. 7). www.atmos-chem-phys.net/7/5727/2007/

Lambe, A. T., Ahern, A. T., Williams, L. R., Slowik, J. G., Wong, J. P. S., Abbatt, J. P. D., Brune, W. H., Ng, N. L., Wright, J. P., Croasdale, D. R., Worsnop, D. R., Davidovits, P., & Onasch, T. B. (2011). Characterization of aerosol photooxidation flow reactors: Heterogeneous oxidation, secondary organic aerosol formation and cloud condensation nuclei activity measurements. *Atmospheric Measurement Techniques*, *4*(3), 445–461. https://doi.org/10.5194/amt-4-445-2011

Li, J., Li, H., Li, K., Chen, Y., Zhang, H., Zhang, X., Wu, Z., Liu, Y., Wang, X., Wang, W., & Ge, M. (2021). Enhanced secondary organic aerosol formation from the photo-oxidation of mixed anthropogenic volatile organic compounds. *Atmospheric Chemistry and Physics*, *21*(10), 7773–7789. https://doi.org/10.5194/acp-21-7773-2021

Ng, N. L., Chhabra, P. S., Chan, A. W. H., Surratt, J. D., Kroll, J. H., Kwan, A. J., Mccabe, D. C., Wennberg, P. O., Sorooshian, A., Murphy, S. M., Dalleska, N. F., Flagan, R. C., & Seinfeld, J. H. (2007a). Atmospheric Chemistry and Physics Effect of NO x level on secondary organic aerosol (SOA) formation from the photooxidation of terpenes. In *Atmos. Chem. Phys* (Vol. 7). www.atmos-chem-phys.net/7/5159/2007/

Ng, N. L., Kroll, J. H., Chan, A. W. H., Chhabra, P. S., Flagan, R. C., & Seinfeld, J. H. (2007b). Secondary organic aerosol formation from m-xylene, toluene, and benzene. In *Atmos. Chem. Phys* (Vol. 7). www.atmos-chem-phys.net/7/3909/2007/

Saiz-Lopez, A., Borge, R., Notario, A., Adame, J. A., Paz, D. D. La, Querol, X., Artíñano, B., Gómez-Moreno, F. J., & Cuevas, C. A. (2017). Unexpected increase in the oxidation capacity of the urban atmosphere of Madrid, Spain. *Scientific Reports*, *7*. https://doi.org/10.1038/srep45956

Sarrafzadeh, M., Wildt, J., Pullinen, I., Springer, M., Kleist, E., Tillmann, R., Schmitt, S. H., Wu, C., Mentel, T. F., Zhao, D., Hastie, D. R., & Kiendler-Scharr, A. (2016). Impact of NOx and OH on secondary organic aerosol formation from β-pinene photooxidation. *Atmospheric Chemistry and Physics*, *16*(17), 11237–11248. https://doi.org/10.5194/acp-16-11237-2016

Song, M., Zhang, C., Wu, H., Mu, Y., Ma, Z., Zhang, Y., Liu, J., & Li, X. (2019). The influence of OH concentration on SOA formation from isoprene photooxidation. *Science of the Total Environment*, *650*, 951–957. https://doi.org/10.1016/j.scitotenv.2018.09.084

Surratt, J. D., Murphy, S. M., Kroll, J. H., Ng, N. L., Hildebrandt, L., Sorooshian, A., Szmigielski, R., Vermeylen, R., Maenhaut, W., Claeys, M., Flagan, R. C., & Seinfeld, J.

H. (2006). Chemical composition of secondary organic aerosol formed from the photooxidation of isoprene. *Journal of Physical Chemistry A*, *110*(31), 9665–9690. https://doi.org/10.1021/jp061734m

Tchepel, O., Costa, A. M., Martins, H., Ferreira, J., Monteiro, A., Miranda, A. I., & Borrego, C. (2010). Determination of background concentrations for air quality models using spectral analysis and filtering of monitoring data. *Atmospheric Environment*, *44*(1), 106–114. https://doi.org/10.1016/j.atmosenv.2009.08.038

Zou, Z., Chen, Q., Xia, M., Yuan, Q., Chen, Y., Wang, Y., Xiong, E., Wang, Z., & Wang, T. (2023). OH measurements in the coastal atmosphere of South China: Possible missing OH sinks in aged air masses. *Atmospheric Chemistry and Physics*, *23*(12), 7057–7074. https://doi.org/10.5194/acp-23-7057-2023